# DISSECTING ATTENTION AND MLP ROLES: A STUDY OF DOMAIN SPECIALIZATION IN LLMS

## ABSTRACT

Large language models (LLMs) perform well across diverse domains such as programming, medicine, and law, yet it remains unclear how domain information is represented and distributed within their internal mechanisms. A key open question is the *division of labor* between the Transformer's core components: self-attention and MLP layers. We address this question through a mechanistic study that dissects their roles by integrating three complementary analyses: representation separability via probes, parameter change under adaptation, and causal effects from activation swaps. Across six domains and multiple models, we find that both Attention and MLP layers encode domain information, but in systematically different ways. We find that attention layers concentrate domain information in localized 'hotspots' (high variance across depth), while MLP layers distribute it uniformly. During fine-tuning, MLPs absorb 2-3× larger parameter updates, yet causal interventions reveal that specific mid-depth attention layers (e.g., layers 13-15) directionally steer domain predictions, while MLP interventions disrupt computation without directional control. These three lenses jointly support a coherent functional picture: MLP layers serve as the primary workbenches for domain-specific computation, while a small subset of attention layers act as high-gain steering points that route domain identity. Finally, we show a proof-of-concept parameter-efficient adaptation setup where tuning only the layers highlighted by our analysis matches full-model fine-tuning on domain benchmarks, illustrating the practical potential of mechanistically informed PEFT.

## 1 INTRODUCTION

Large Language Models (LLMs) master diverse domains, yet the internal mechanisms governing this domain representation remain an open question. *What is the division of labor between the Transformer's core components – the self-attention and the MLP layers?* In this paper, we address these questions through a causal, layer-level analysis and propose a functional specialization that holds across models and domains.

The field of mechanistic interpretability has developed powerful methods for such analysis, progressing from correlational analysis to causal interventions. Initial probing (Alain & Bengio, 2018; Tenney et al., 2019) analyses used simple neural classifiers to differentiate the outputs of a layer for varied inputs. A highly separable representation of domain identity, for example, would imply that a component contains domain-specific information. The contribution can be quantified by calculating the level of separation in the higher-dimensional space through separability scores, like v-usable information (Ethayarajh et al., 2022) , Xu et al. (2020) , Ju et al. (2024b) Fisher separability Fisher (1936), maximum mean discrepancyGretton et al. (2008) etc. Although these methods can show where information separates, but not if or how the model uses it for downstream tasks.

Subsequently, the focus shifted towards establishing causality by reverse-engineering the circuits (Elhage et al., 2021) for specific behaviors, through methods like activation patching (Meng et al., 2023a) Wang et al. (2022) and zero-out testing (Dai et al., 2021). This research has yielded an important result: MLP layers have been characterized as the primary locus of holding factual knowledge. Concurrently, attention mechanisms are understood as routers, moving and aligning information throughout the context, enabling capabilities like in-context learning (Olsson et al., 2022).

A parallel line of evidence comes from studying parameter adaptation. Research on techniques like Low-Rank Adaptation (LoRA) has shown that model behavior can be improved for targeted adaptation by modifying only a small subset of weights (Hu et al., 2022) Zhang et al. (2023). However, a critical gap remains. These three powerful lenses—representational, causal, and adaptational—have largely been applied in isolation and to micro-scale tasks (e.g., factual recall, syntactic phenomena). It is unknown whether the "attention-as-router, MLP-as-compute" principle scales to govern how models handle high-level, abstract domains like programming or medicine. Furthermore, no existing framework exists to synthesize these three orthogonal sources of evidence into a single, coherent map of a model's functional architecture.

**Our perspective.** We study *domain handling* in LLMs by jointly applying three lenses to the same models and domains. Concretely, we ask the following questions, at the level of individual Transformer blocks:

1. **Where is domain identity represented?** We measure how separable domain labels are from layer activations using Fisher ratio and kernel MMD.

2. **Where does domain adaptation write new computation?** We quantify layer-wise parameter changes under domain-specific LoRA fine-tuning.

3. **Which layers can steer domain-sensitive behavior under intervention?** We perform activation swaps between matched domain prompts and measure both disruption and directional shifts in predictions.

Here, we qualify that domain control is complex and distributed; no single component type exclusively handles all aspects. Our analysis reveals relative differences in component contributions rather than absolute divisions. In summary, our contributions are:

- **Representational separability is shared, but distributed differently.** After excluding trivially input-driven and logit-driven layers, both Attention and MLP components broadly encode domain identity. However, separability across depth has markedly higher variance in Attention than in MLP: a few attention layers form sharp "hotspots" of domain-specific separability, whereas MLP layers encode domain information more uniformly.

- **Evidence for a Scaled Division of Labor in Abstract Tasks**: We provide direct evidence that the "attention as router, MLP as memory" principle, previously observed in low-level factual tasks, also governs how models handle high-level, abstract capabilities like domain control. This suggests it is a fundamental organizing principle of the Transformer architecture.

- **Demonstration of Mechanistically-Informed, Parameter-Efficient Fine-Tuning as proof of concept**: We show that our mechanistic map can be used to have direct practical utility. The models' performance matches full-model fine-tuning on our domain benchmarks, illustrating the practical potential of mechanistically guided PEFT.

## 2 PROPOSED METHODOLOGY

Our work examines the roles of attention and MLP components across layers through/via three perspectives: representational patterns (Probing analysis), parameter changes (Fine tuning analysis), and causal interventions.

### 2.1 PROBING ANALYSIS

The objective of this experiment is to identify which layers contain the most linearly separable information about domain identity. Classical classification accuracy saturated at around 100% across all layers, providing insufficient discriminative power to determine where domain information is most concentrated. We instead quantify the degree of separability using distributional metrics. A high degree of separability indicates that a layer's activations serve as a strong signal for the domain, a necessary condition for a component involved in routing or high-level control.

To quantify where domain identity is explicitly represented, we compute pairwise separability between domains for each layer and component using two complementary statistics: a scalar Fisher ratio Fisher (1936) and RBF-kernel Maximum Mean Discrepancy (MMD) Gretton et al. (2008). Let

$X \in \mathbb{R}^{N \times D}$ be pooled activations for a given (layer, component) and $y \in \{1, \ldots, K\}^N$ the domain labels. Denote by $X_i$ the rows of $X$ with label $i$, $N_i = |X_i|$, and $\mu_i = \frac{1}{N_i} \sum_{x \in X_i} x$.

**Fisher** : We report the scalar Fisher score between domains $i$ and $j$:

$$\text{Fisher}_{ij} = \frac{\|\mu_i - \mu_j\|^2}{\sum_{d=1}^{D} \text{Var}(X_{i,\cdot d}) + \sum_{d=1}^{D} \text{Var}(X_{j,\cdot d}) + \varepsilon},$$

with $\varepsilon = 10^{-6}$ for numerical stability. This ratio is high when domain means are well-separated relative to within-domain variance, indicating linear discriminability.

**MMD (RBF).** Using an RBF kernel $k_\gamma(x, x') = \exp(-\gamma \|x - x'\|^2)$ we compute

$$\text{MMD}^2_{k_\gamma}(X_i, X_j) = \frac{1}{N_i^2} \sum_{a,b \in X_i} k_\gamma + \frac{1}{N_j^2} \sum_{a,b \in X_j} k_\gamma - \frac{2}{N_i N_j} \sum_{a \in X_i} \sum_{b \in X_j} k_\gamma,$$

and report $\text{MMD}_{ij} = \sqrt{\max(0, \text{MMD}^2)}$. The kernel bandwidth $\gamma$ is set by the median heuristic on pairwise distances.

Activations are extracted by registered forward hooks at two probe points per block: post-attention and post-MLP (before residual addition). (for details on pipeline see Appendix A.2). We display only Fisher and MMD scores because they capture complementary linear (mean-vs-variance) and nonlinear (higher-moment) distributional differences and provide the clearest layer-wise differentiation in our experiments. Rather than exhaustively reporting all $\binom{K}{2}$ pairwise scores, we compute **a 1-vs-all** statistic for each domain. For a domain $D_i$, activations from $D_i$ are compared against the pooled activations from all other domains $\bigcup_{j \neq i} D_j$. This yields a per-layer, per-component separability score $S_{i,\ell}$ indicating how well layer $\ell$ distinguishes $D_i$ from the rest of the corpus. To compare components on the same scale, we z-normalize scores across layers for each domain. We observed strong separability in the first and last layers, but further analysis (Appendix E.1) shows that these peaks largely reflect input-distribution differences and logit effects, rather than internal steering. To avoid these confounds and focus on internal specialization, we exclude these layers from summary statistics such as variance and maximums in the main text.

## 2.2 Fine-tuning analysis

Probing identifies where domain identity is *separated* in activations; the complementary question is where parameters undergo adaptation. We answer this by measuring per-layer parameter updates under fine-tuning and by testing whether the layers that change most are also the layers that suffice for adaptation.

We use LoRA-style fine-tuning for targeted, parameter-efficient adaptation. For a dense weight $W \in \mathbb{R}^{n \times m}$ at layer $\ell$ the adapted weight is $W + \Delta W_\ell$ with $\Delta W_\ell = \frac{\alpha}{r} B_\ell A_\ell$ where $A_\ell \in \mathbb{R}^{r \times m}$, $B_\ell \in \mathbb{R}^{n \times r}$, $r$ is the adapter rank and $\alpha$ is a scalar scaling. We summarize a layer's adaptivity by the Frobenius norm of the effective update

$$S_\ell = \|\Delta W_\ell\|_F,$$

and aggregate multiple adapter tensors that belong to the same Transformer block by summation: $S_\ell^{\text{block}} = \sum_{t \in T_\ell} \|\Delta W_t\|_F$. A high $S_\ell$ indicates that the parameters in $\ell$ layer are a primary site for storing new, domain-specific computation learned during adaptation Gupta et al. (2025).

We run three fine-tuning regimes: (i) full-model fine-tuning (baseline), (ii) LoRA targeted only to attention projection matrices (e.g., $q, k, v, o$ per block), and (iii) LoRA targeted only to MLP projection matrices (e.g., gate/up/down). To validate the utility of our layer map, we additionally fine-tune only the top-3 and bottom-3 layers ranked by separability. Crucially, to ensure robust adaptation and prevent catastrophic forgetting, we augment the training data with a set of generic prompts and apply loss masking so that gradients are backpropagated only from the model's generated responses, not the input instructions. For domain perplexity evaluation, we additionally fine-tune only the top 1 and top 3 layers under each of these regimes. All fine-tune runs use fixed hyperparameters (epochs, learning rate, batch size, LoRA rank) and multiple random seeds to enable statistical comparison. (See Appendix D.1)

## 2.3 CAUSAL ACTIVATION SWAPPING

Probing and fine-tuning establish where domain information is present and where the optimizer writes it; to show that a layer's activations actually *cause* domain-directed generation, we perform activation swapping. The experiment asks: if we transplant the hidden state from a donor prompt in domain $D_b$ into a recipient prompt in domain $D_a$, does the model's next-token distribution shift toward $D_b$?

To rigorously isolate domain routing from generation complexity, we replace the open-ended code generation task with a controlled **Domain Classification** task. We construct matched prompt pairs using the following template:

```
Below are two sets of keywords that you need to classify into two domains.
(A): [List of n representative tokens from domain A]
(B): [List of n representative tokens from domain B]
Which set is domain X? Answer: Option (
```

From this template, we define the recipient input $x_a$ as the "correct" prompt where the queried domain $X$ corresponds to list (A). Conversely, the donor input $x_b$ is a "conflicting" prompt where the queried domain $X$ corresponds to list (B). Ideally, the model predicts "A" for $x_a$. Our goal is to determine if injecting activations from $x_b$ steers the model to predict "B".

We focus our causal analysis on multiple domain pairs (e.g., C++/Python, Medicine/Finance) to ensure generalizability. This setup offers several methodological advantages: (1) structural prompt similarity enables precise matched comparisons, (2) the output space is restricted to binary classification labels (A vs B), providing a clean directional signal, and (3) computational complexity is identical across samples, isolating domain identity from task difficulty. For a chosen layer $\ell$ and the final prompt position $t^\star$, we:

1. run a forward pass on the donor (conflicting) input $x_b$ and save donor activations $a_\ell^{\mathrm{donor}}(t^\star)$;

2. run a forward pass on the recipient (correct) input $x_a$ but, at layer $\ell$ and position $t^\star$, replace the recipient activation with $a_\ell^{\mathrm{donor}}(t^\star)$ and continue inference to obtain the patched distribution $p_{\mathrm{swap}(\ell)}(\cdot \mid x_a)$;

3. repeat across many donor–recipient pairs and average metrics (see E.5).

**Metrics.** We quantify the effect of a swap with two complementary statistics that capture magnitude and directionality.

*(1) KL Divergence.* For a donor input $x_b$ and recipient input $x_a$ we define

$$\mathrm{KL}_{\mathrm{swap}\,\ell} \;=\; \mathbb{E}_{x_a}\big[\,\mathrm{KL}\big(p(\cdot \mid x_a) \,\|\, p_{\mathrm{swap}(\ell)}(\cdot \mid x_a)\big)\,\big],$$

where $p(\cdot \mid x_a)$ is the original next-token distribution and $p_{\mathrm{swap}(\ell)}(\cdot \mid x_a)$ is the patched distribution. $\mathrm{KL}_{\mathrm{swap}\,\ell}$ measures how strongly the swap perturbs the model's predictive distribution at the intervention point.

*(2) Delta bias.* We define the target token sets $S_A = \{\text{"A"}\}$ and $S_B = \{\text{"B"}\}$ corresponding to the options in the prompt. For a prompt $x_a$, let $P(S|x)$ be the probability mass on tokens $S$. Bias toward the donor outcome is $\mathrm{Bias}(x) = P(S_B|x) - P(S_A|x)$ We measure the change due to intervention as

$$\Delta\mathrm{Bias}(D_a \overset{\ell}{\leftarrow} D_b) = \mathbb{E}\Big[\mathrm{Bias}_{swap}(x_a \overset{\ell}{\leftarrow} x_b) - \mathrm{Bias}_{base}(x_a)\Big].$$

Positive values indicate a shift toward the donor label, since bias is computed as the preference of the donor option over the recipient option. For complete details, see Appendix E.4

KL captures whether an intervention meaningfully alters the model's beliefs; the domain-token Shift tests whether the alteration is *directionally* consistent with the donor domain. Together they provide strong, local causal evidence that activations at layer $\ell$ not only correlate with domain identity but can drive domain-appropriate generation when transplanted into another context. The experimental conditions ensure that trivial scale differences do not drive observed effects. For more implementation details, see Experimental Setup E.5

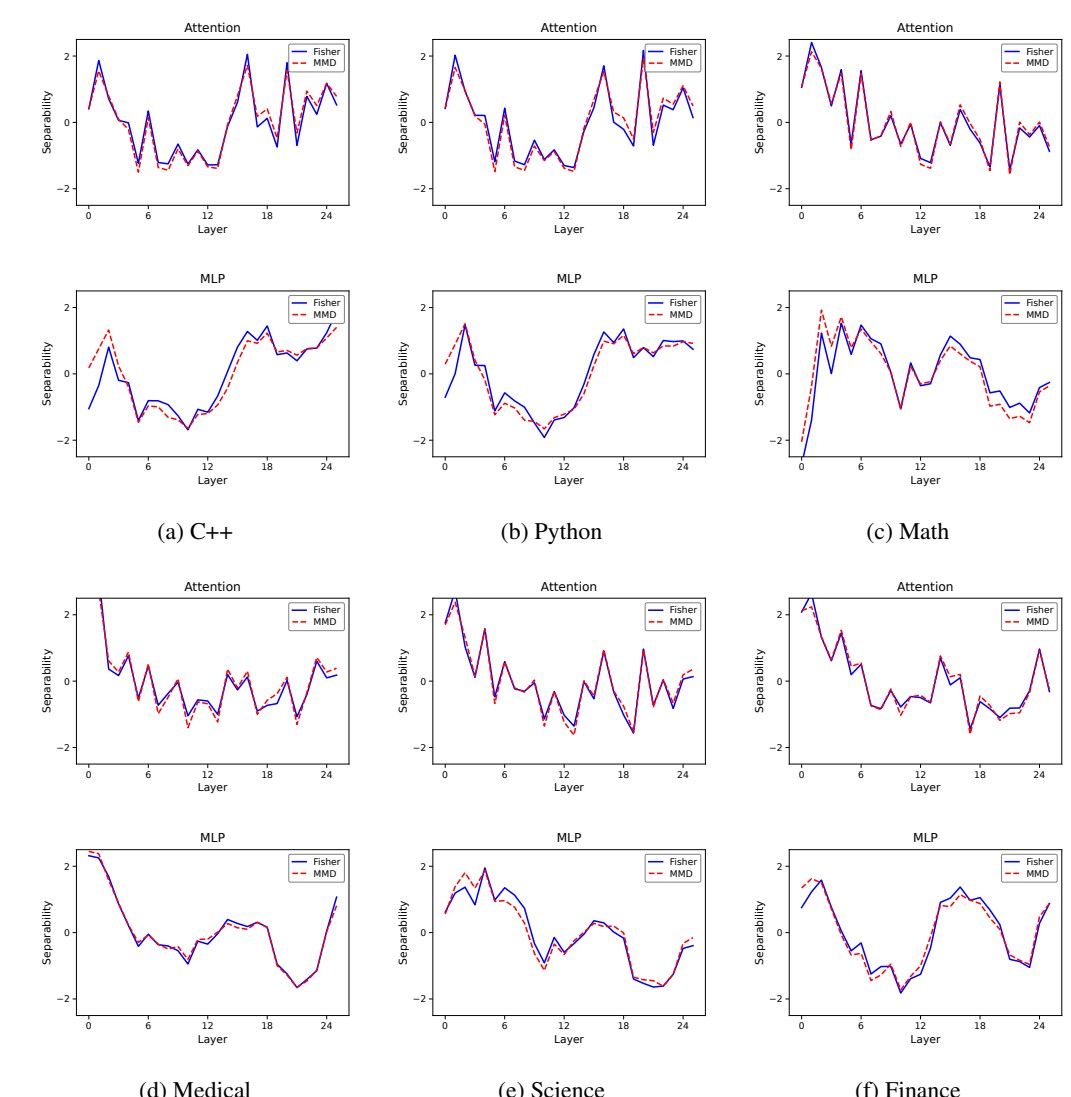

Figure 1: Separability scores across six domains. Each column displays Attention (top) and MLP (bottom) blocks for one domain.

# 3 RESULTS

Our investigation spans six domains: Medicine, Finance, Science, Mathematics, C++, and Python, and on four LLMs: Llama 3.2 3B, Llama 1B, Gemma 3 4B, and Gemma 3 1B (Grattafiori et al., 2024; Team et al., 2025). For more details on datasets used, see Appendix B. The following discussion is for the Llama 3.2 3B model, which consists of 28 layers, each with an MLP head and an attention mechanism. Consistent with our analysis of the 'Hydra effect' (Appendix E.1), we focus the following quantitative results on the internal layers, excluding the immediate embedding and final output layers where representations are dominated by input/output constraints rather than internal processing. For results on other models, see Appendix A.

## 3.1 WHERE DOMAIN KNOWLEDGE IS SEPARATED?

Figure 1 shows the 1-vs-all Fisher and MMD separability traces across layers for six domains, z-normalized to highlight relative variation in depth. Both Attention and MLP components exhibit non-uniform separability: some layers carry markedly stronger domain identity than others. While the

| Domain | Attention | | | | MLP | | | |
| --- | --- | --- | --- | --- | --- | --- | --- | --- |
| | Fisher | | MMD | | Fisher | | MMD | |
| | Max | Std | Max | Std | Max | Std | Max | Std |
| CPP | 1.386 | 0.212 | 0.617 | 0.049 | 1.236 | 0.093 | 0.598 | 0.019 |
| Python | 1.359 | 0.202 | 0.615 | 0.048 | 1.038 | 0.067 | 0.592 | 0.020 |
| Medical | 1.532 | 0.218 | 0.657 | 0.037 | 1.392 | 0.110 | 0.664 | 0.017 |
| Science | 1.281 | 0.169 | 0.630 | 0.033 | 1.049 | 0.071 | 0.606 | 0.013 |
| Math | 1.356 | 0.191 | 0.639 | 0.035 | 1.062 | 0.060 | 0.613 | 0.011 |
| Finance | 2.307 | 0.323 | 0.717 | 0.031 | 1.987 | 0.133 | 0.714 | 0.010 |

Table 1: Maximum Value and standard deviation of 1-vs-all separability scores for Attention and MLP layers across six domains. Higher values indicate greater domain specificity and localization for that component.

overall trends are similar, the precise peaks do not fully coincide between Attention and MLP. This suggests that both components participate in domain representation, but their strongest contributions arise at slightly different depths.

After z-score normalization, Fisher and MMD traces nearly completely overlap across layers. This indicates that both linear mean-based separation (Fisher) and higher-moment distributional divergence (MMD) identify the same loci of domain information. Thus, the observed peaks are not artifacts of a particular separability metric, but reflect genuine structural patterns in the residual stream.

To compare components, Table 1 reports the maximum and standard deviation of 1-vs-all separability scores across layers. A clear pattern emerges. While both components possess domain information, the *distribution* of this information differs fundamentally. MLP layers exhibit consistently low standard deviation across all domains (e.g., C++ Fisher Std = 0.093), implying that domain-specific features are distributed relatively uniformly across depth. In contrast, Attention layers show significantly higher variance (e.g., C++ Fisher Std = 0.212), indicating that domain identity is not uniform but highly concentrated at specific 'hotspot' layers. The maximum separability reinforces this distinction. For all 6 domains, the maximum Fisher and MMD scores are higher for Attention layers than for MLP layers. For the C++ domain, the peak Attention separability (1.386) exceeds the peak MLP separability (1.236), but more importantly, the variance is over $2\times$ higher in Attention. This indicates that while domain information is generally available, it becomes highly concentrated at specific bottleneck layers within the Attention mechanism.

## 3.2 ADAPTATIONAL ANALYSIS POINTS TO MLP LAYERS

While probing analysis suggests concentrated signals in attention layers, adaptational analysis reveals a different picture. Figure 2 plots the average normalized weight change ($\|\Delta W\|/\|W\|$) per layer for three LoRA fine-tuning regimes: targeting the full model, only MLP components, or only attention components.

The magnitude of weight change in MLP-only fine-tuning is substantially and consistently higher than in attention-only fine-tuning. This indicates that MLP layers are the primary locus where new, domain-specific computation is written during adaptation to a specific domain. The results are unambiguous across all six domains, persisting even when using loss masking and generic prompt augmentation to prevent overfitting. This implies that while attention layers had concentrated signals due to higher peaks of separability in specific layers, adapting to a new dataset consistently relies on modifying the MLP layers more, proposing that domain-specific knowledge is *stored* in the latter.

## 3.3 VALIDATING THE PROPOSED LAYER MAP VIA TARGETED FINE-TUNING

Before performing causal interventions, we first seek to validate the practical utility of our proposed layer map. If the layers, either those with the largest parameter deltas (primarily MLPs) or those with the most separable representations (peak attention layers), are indeed the most important for adaptation, then fine-tuning only these layers should achieve satisfactory results in comparison to fine-tuning the entire model. We test this hypothesis by fine-tuning only the top-3 and bottom-3 layers

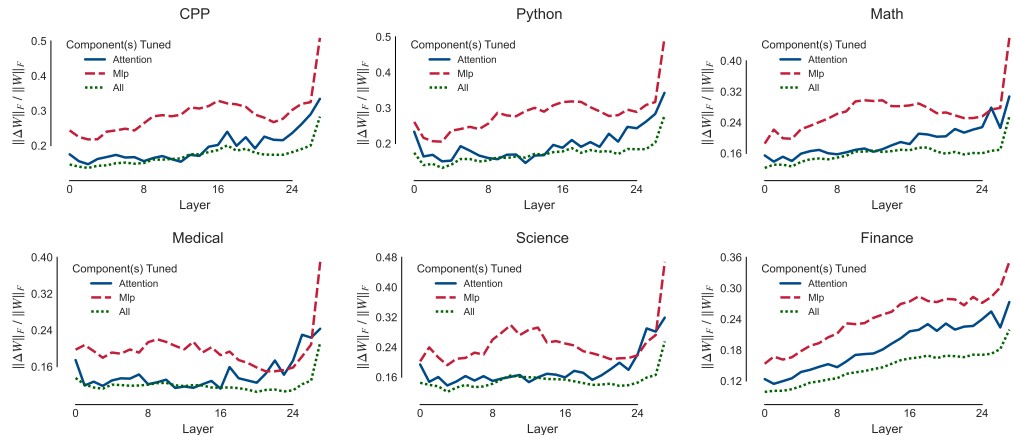

Figure 2: Change in the weights on Lora-based fine-tuning, separately on (1) Entire model, (2) Only Attention Layers, and (3) Only MLP Layers

(for both MLP and Attention) ranked by their separability scores and comparing their performance on the respective domain's specific perplexity task against fine-tuning the full model. For every domain, the domain perplexity was devised using a benchmark evaluation method, normalized between 0 and 1. Details of domain-specific evaluation are mentioned in Appendix C.2.

Interestingly, the results in Table 2 are even better than expected. Targeted fine-tuning of just the top-3 layers achieves domain-specific performance that is comparable to, and in some cases exceeds, that of fine-tuning the entire model, despite using a fraction of the parameters. For more insights, refer to Appendix C.1. The dataset used for fine-tuning had around 5000-7000 samples, as discussed in B.

It is important to note that due to the small scale of the models and limited fine-tuning data, fine-tuning can suffer from some forgetting of general capabilities. However, the relative performance gain across all fine-tuned results demonstrates that our layer importance map successfully identifies the most critical components for specialization. We emphasize that we do not claim targeted PEFT is superior to full-model tuning. Instead, these results serve as a proof-of-concept indicating that mechanistically-guided selection of layers can enable efficient adaptation, highlighting the potential of interpretability to inform practical fine-tuning strategies.

|  | PT | Full Fine Tuning | Bottom-3 MLP | Bottom-3 Attn | Top-3 MLP | Top-3 Attn |
|---|---|---|---|---|---|---|
| Math | 0.07 | 0.02 | 0.08 | 0.07 | **0.12** | 0.03 |
| Science | 0.88 | 0.88 | 0.87 | 0.88 | **0.88** | 0.86 |
| CPP | 0.31 | 0.41 | 0.30 | 0.34 | 0.39 | **0.41** |
| Python | 0.73 | 0.69 | 0.68 | 0.65 | 0.71 | **0.73** |
| Finance | 0.94 | 0.95 | 0.91 | 0.91 | 0.93 | **0.95** |
| Medical | 0.67 | 0.68 | 0.66 | 0.67 | 0.67 | **0.68** |

Table 2: Performance of Llama-3.2-3B across domains on that domain-perplexity metric (normalized between 0 and 1). PT stands for pre-trained model. All the other column names resemble the components fine-tuned during adaptation.

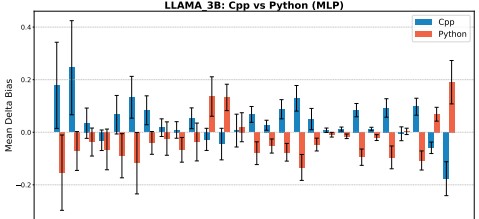 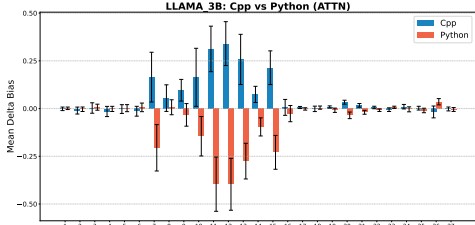

Figure 3: **Causal intervention results across all layers for Llama-3.2-3B.** Delta Bias when swapping activations between C++ and Python prompts using our domain-classification task (Section 2.3). X-axis: layer index (0-27). Y-axis: Delta bias. Swapping attention activations produces large, positive shifts at specific mid-depth layers (e.g., 13-15, 23-25), indicating sparse routing hotspots. Error bands show standard deviation over 200 prompt pairs. Full results for all domain pairs in Appendix E.5.

### 3.4 CAUSAL SWAPPING REVEALS ATTENTION AS DOMAIN ROUTER

Probing identified separable domain representations, and adaptation revealed MLPs as the primary locus of parameter change. To test which components actually *cause* domain-directed behavior, we use activation swapping in our Domain Classification task.

We measure the overall magnitude of the intervention's effect using KL divergence as shown in figure 3. For components in layers with high Fisher separability, swapping activations from either an attention block or an MLP block induces a significant relative perturbation in the next-token distribution, resulting in high KL divergence. This confirms that both components in these layers are computationally active and influential on the final output. Conversely, interventions on components in low-Fisher layers produce a negligible KL divergence, confirming that the effect is localized to the information-rich parts of the network. Early layers occasionally exhibit high Fisher but low causal effect (e.g., Attention layer-2), suggesting the occurrence of "hydra" effect McGrath et al. (2023) (Discussion E.1) here.

However, a disruptive effect does not imply directional control. To test if a layer steers the output towards a specific domain, we measure the shift in probability mass towards the donor option token (i.e., the label corresponding to the conflicting domain). Crucially, we observe that this steering capability is highly specialized, and can be seen as distinct routing hotspots within specific mid-depth layers. In these concentrated points, swapping activations from a conflicting donor into a correct prompt reliably shifts the prediction toward the conflicting label. This provides direct causal evidence that these specific attention layers are not just active, but are providing a sparse, specific steering signal for domain identity. In contrast, swapping the output of a high-Fisher MLP layer does not produce a consistent directional shift. While the intervention is disruptive (high KL), the effect on the target label probability (Delta Bias) is centered around zero across all layers, albeit with high variance. This suggests that while the MLP is performing critical, domain-relevant computations, it is not the source of the high-level control signal that dictates "the answer is Option B."

On bringing together these observations, we can conclude that the MLP layers change most during fine-tuning because they are the computational workbenches where domain-specific knowledge is implemented. Intervening on them is disruptive because it interrupts this computation. However, it is the sparse set of peak attention layers that act as the causal routers. Their activations, though less plastic during fine-tuning, carry the high-level steering signal that *directs* the downstream computational machinery of the MLPs.

## 4 DISCUSSION

Our investigation began with a foundational question: how does a monolithic network manage distinct domains? By analyzing the three lenses as proposed, we have moved beyond simple observation to a causal, mechanistic explanation. Our results resolve the apparent contradiction between representational and adaptational analyses, revealing a clear and consistent division of labor between the Transformer's core components. Here, we synthesize these findings, discuss their implications for the field, and outline the limitations of our work to chart a path for future research.

**Transferability across models.** Our findings are not confined to a single checkpoint. We executed all analyses on LLaMA-1B, LLaMA-3.2B, and Gemma 3-1B/4B (See A). We further validated our primary causal claims on the larger Llama-2-7B model to ensure scalability. The overall pattern holds: specific attention layers exhibit localized, high-separability peaks that act as causal routers, while MLP layers accumulate the bulk of adaptation updates. Interestingly, Gemma models display sharper, more localized separability in causal swap experiments, with a single attention layer causing large directional effects. This acute localization of causal influence suggests a more specialized, hub-like routing mechanism within Gemma's architecture, suggesting that architectural choices, such as logit soft-capping or normalization, may influence the concentration of domain representation. These findings highlight the need to explore how such architectural decisions affect causal control and domain adaptation, offering a promising direction for future research.

**A coherent mechanistic picture.** Taken together, our three experiments point to a consistent proposition. First, probing reveals a structural distinction: while both components encode domain identity, MLP layers exhibit low variance across depth, implying a distributed representation. In contrast, Attention layers exhibit high variance with sharp hotspots of separability. Second, adaptation analysis confirms that MLPs function as the primary workbenches; even under robust fine-tuning conditions preventing overfitting, they absorb the vast majority of parameter updates. Third, causal interventions explain this structure. Swapping activations in the high-variance Attention layers provides a clean *steering* signal , reliably flipping the model's decision in classification tasks. In contrast, swapping MLP activations causes *disruption* without directional steering. In the domain level of abstraction, attention acts as the router, steering domain identity, while MLPs implement the downstream computations that realize domain-specific behavior.

**Implications.** This proposal has two important implications. First, it provides a layer-level map of where to look for domain control in Transformers, guiding mechanistic interpretability beyond micro-circuits to higher-level behaviors. Second, it has practical value: our targeted fine-tuning experiments serve as a proof-of-concept, demonstrating that a small set of components identified by our map suffices to replicate full-model domain tuning. This highlights the potential for mechanistically-grounded strategies to enable more efficient model adaptation.

**Limitations and caveats.** Our study has several limitations. (i) To isolate steering from generation, we relied on a controlled Classification task; however, complex open-ended generation may involve more distributed control signals that single-point swaps cannot fully capture. (ii) We adopt a 1-vs-all separability framework, which simplifies analysis but may collapse informative pairwise distinctions between domains. (iii) Our models are relatively small and fine-tuned on modest datasets; while we validated causal effects on 7B models, emergent behaviors in 70B+ scale models remain an open question. (iv) Early-layer separability peaks (e.g., A2) did not always yield causal effects, consistent with the hydra effect, where distributed signals do not translate into single-point steering handles. (v) Finally, our causal swaps measure immediate next-token shifts; long-horizon effects and global coherence remain to be tested.

**Future directions.** These caveats suggest clear paths forward. Future work should extend our work to even larger and more diverse models, refine domain prompts beyond code pairs, and analyze per-head specialization within the identified router layers. A natural next step is to connect layer-level maps to explicit circuit motifs, integrating coarse-grained and fine-grained mechanistic interpretability. On the practical side, our study could be used to guide efficient domain adaptation or controlled editing, narrowing the intervention space to the components that matter most. Finally, a critical direction is to investigate how this implicit division of labor maps onto architectures with explicit routing, such as Mixture-of-Experts (MoE).

## 5 RELATED WORK

**Representation analysis** : The use of simple linear classifiers, or probes, to correlate internal activations with linguistic properties marked an early effort to map knowledge in neural networks (Alain & Bengio, 2018; Tenney et al., 2019). This method was quickly refined in response to critiques that high accuracy does not guarantee task-relevance, leading to the development of control methods and more sophisticated layer-wise analyses of information gain (Hewitt & Liang, 2019; Ravichander et al., 2020; Kunz & Kuhlmann, 2022). Applied to contemporary LLMs, these refined techniques

have revealed clear knowledge hierarchies: the "Concept Depth" hypothesis posits that complex concepts are processed in deeper layers (Jin et al., 2024), while abstract traits like personality are localized to the middle-to-upper layers (Ju et al., 2024a). The search for greater precision has led to techniques like sparse probing for isolating the specific neurons responsible for a concept (Gurnee et al., 2023), and has connected analysis to action by using probe results to guide targeted edits on model behavior (Li et al., 2024).

Critically, **targeted fine-tuning of top-3 layers often matches or exceeds full-model performance**, suggesting that selective adaptation to mechanistically-identified layers can mitigate overfitting by constraining the parameter space while preserving domain-relevant updates. However, we emphasize that these results serve primarily as proof-of-concept for the utility of mechanistic layer selection, not as a claim that our current PEFT approach is superior to full-model training at scale. **Causal interventions**: To move from correlation to causation, a central method is activation patching: a family of techniques that swap activations between inputs to measure their causal effect (Vig et al., 2020; Geiger et al., 2021; Heimersheim & Nanda, 2024). Its application to model editing began with locating and updating single facts via ROME (Meng et al., 2022), a process later scaled to thousands of facts with MEMIT (Meng et al., 2023b) and made more efficient by SaLEM (Mishra et al., 2024). The scope of such causal analysis has since expanded beyond discrete facts, used to map the locality of categorical knowledge (Burger et al., 2024) and to reverse-engineer entire computational circuits 'in the wild' (Wang et al., 2022).

**Functional Specialization of Transformer Components** : Causal analysis reveals a functional specialization between a transformer's primary sub-layers. MLP layers are established as key-value memories that store factual knowledge (Geva et al., 2021), a view substantiated by causal editing (Meng et al., 2022) and shown to hold in multilingual contexts (Fierro et al., 2023). Conversely, attention mechanisms act as dynamic routers, moving information through the residual stream (Elhage et al., 2021; Olsson et al., 2022). This simple dichotomy has evolved into a more nuanced view of integrated knowledge circuits, with work formalizing how attention filters information for MLPs to store (Xu & Chen, 2023) and detailing direct Attention-MLP interactions (Yao et al., 2024; Neo et al., 2024).

**Parameter-Efficient Fine-Tuning as a Locus of Knowledge**: A parallel line of research frames Parameter-Efficient Fine-Tuning (PEFT) as a mechanistic diagnostic. While foundational methods like Adapter-tuning (Houlsby et al., 2019) and LoRA (Hu et al., 2022) were developed for engineering efficiency, why and where they work has deep mechanistic implications. Analyses suggest LoRA learns low-rank updates that mimic full fine-tuning (Zhang et al., 2023), and critically, that the efficacy of these updates is highly dependent on their layer-wise placement (An et al., 2024; He et al., 2022). This localization principle is further exemplified by methods like LoFiT, which use interpretability to identify and then fine-tune only a sparse subset of task-critical attention heads (Yin et al., 2024).

## 6 CONCLUSION

We demonstrate a clear division of labor in Transformers at the high-level scale of complex, real-world domains: attention layers route domain identity, while MLP layers store domain-specific knowledge. This work establishes that the "router-compute" principle—previously observed in low-level tasks—organizes high-level domain specialization across programming, medicine, and other complex domains. By triangulating probing, adaptation, and causal interventions, we provide a definitive functional map: attention layers serve as domain routers that causally steer model behavior, while MLP layers act as domain-specific computational units. This architectural insight provides a blueprint for more interpretable and efficient model adaptation, advancing our understanding of how large language models master diverse capabilities.

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

APPENDIX

# A RESULTS ON OTHER MODELS

## A.1 FINE TUNING ANALYSIS

**Stage 1: Comprehensive Adaptational Mapping.** The initial stage conducted a broad, component-wise analysis for each of the six domains independently. To map the division of labor between Transformer components, we applied LoRA adapters under three distinct regimes:

- **Attention-Only**: LoRA was applied exclusively to the attention projection matrices (q_proj, k_proj, v_proj, o_proj) in every layer.
- **MLP-Only**: LoRA was applied exclusively to the MLP projection matrices (gate_proj, up_proj, down_proj) in every layer.
- **Full Model (All)**: LoRA was applied to all attention and MLP components simultaneously, establishing a baseline for unconstrained, full-model adaptation.

The primary objective of this stage was to quantify the magnitude of parameter updates for each component $c \in \{\text{Attn, MLP}\}$ at each layer $\ell$, measured by the Frobenius norm of the effective weight change, $S_{\ell,c} = \|\Delta W_{\ell,c}\|_F$. The results from this analysis provide the data for the adaptational plots in the main paper (Figure 2) and this appendix.

### STAGE 1 RESULTS FOR OTHER MODELS

The adaptational patterns observed in the Llama 3.2 3B model hold consistently across other model families and sizes, as shown below.

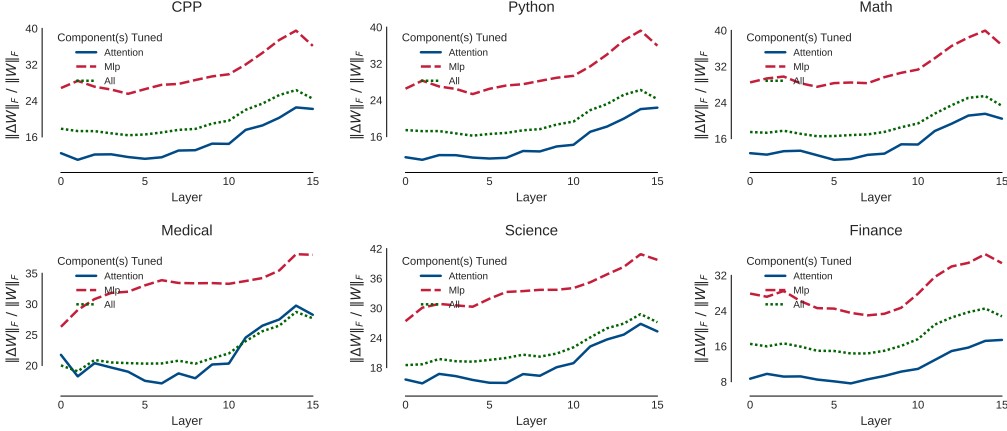

Figure 4: Layer-wise magnitude of parameter updates ($S_\ell$) for **Llama 3.2 3B** under three LoRA fine-tuning regimes across six domains.

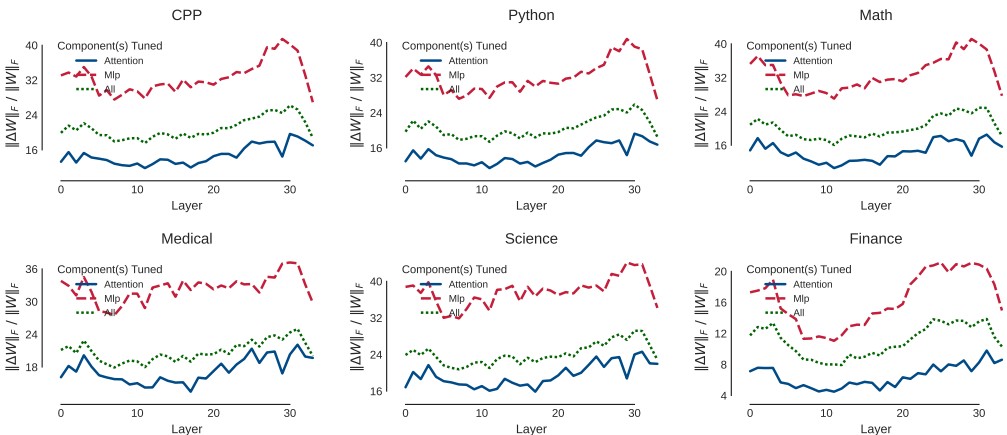

Figure 5: Layer-wise magnitude of parameter updates $(S_\ell)$ for **Gemma 3 4B** under three LoRA fine-tuning regimes across six domains.

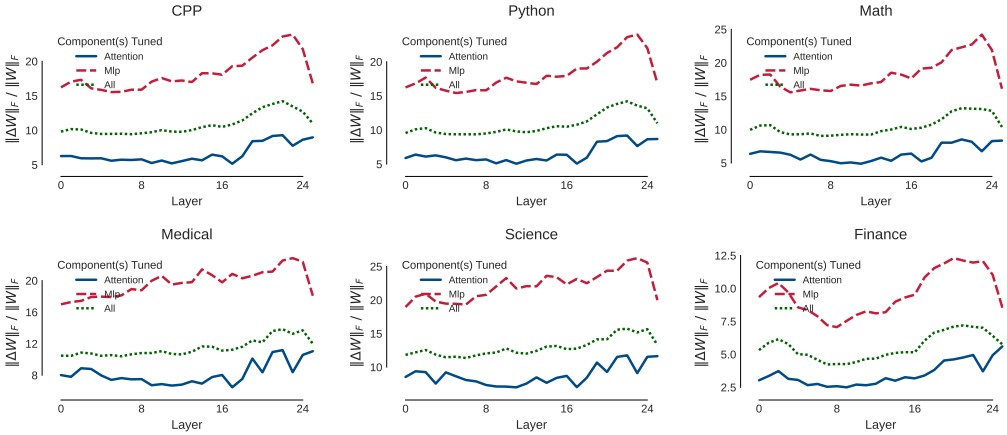

Figure 6: Layer-wise magnitude of parameter updates $(S_\ell)$ for **Gemma 3 1B** under three LoRA fine-tuning regimes across six domains.

ADAPTATIONAL NORM ANALYSIS

To dissect the dynamics of targeted adaptation, we compare the norms of LoRA weight updates ($\|\Delta W_\ell\|_F$) for the top-3 most-adapted layers across three analytical contexts. The summary tables aggregate these norms to reveal overarching patterns.

- **Avg. Full Run Norm**: The average norm of a component group (e.g., Top-3 MLPs) from the Stage 1 "Full Model" regime, where all layers were adapted on a single domain. This represents the baseline update magnitude in an unconstrained setting.

- **Avg. Ensemble Norm**: The average norm of a component group from a Stage 2 "Ensemble Tuning" run, where *only* those specific components (e.g., only the Top-3 MLP layers) were adapted. This measures the update magnitude under targeted, multi-component fine-tuning.

- **Top Solo Run Norm**: The norm of the single highest-ranking component from a Stage 2 "Soloist Tuning" run, where it was the *only* component adapted in the entire model. This quantifies a component's adaptational capacity in complete isolation.

Table 3: Aggregated LoRA weight update norms for the Llama 3.2 3B model across all domains.

| Domain | Component Group | Avg. Full Run Norm | Avg. Ensemble Norm | Top Solo Run Norm |
|---|---|---|---|---|
| CPP | Top-3 MLP Components (Avg.) | $1.019 \times 10^2$ | $1.287 \times 10^2$ | $1.651 \times 10^2$ |
| | Top-3 Attn Components (Avg.) | $7.042 \times 10^1$ | $9.357 \times 10^1$ | $1.149 \times 10^2$ |
| Finance | Top-3 MLP Components (Avg.) | $9.463 \times 10^1$ | $6.990 \times 10^1$ | $9.945 \times 10^1$ |
| | Top-3 Attn Components (Avg.) | $5.464 \times 10^1$ | $4.687 \times 10^1$ | $6.056 \times 10^1$ |
| Math | Top-3 MLP Components (Avg.) | $1.007 \times 10^2$ | $1.360 \times 10^2$ | $1.676 \times 10^2$ |
| | Top-3 Attn Components (Avg.) | $6.580 \times 10^1$ | $8.377 \times 10^1$ | $9.766 \times 10^1$ |
| Medical | Top-3 MLP Components (Avg.) | $9.545 \times 10^1$ | $1.239 \times 10^2$ | $1.560 \times 10^2$ |
| | Top-3 Attn Components (Avg.) | $9.134 \times 10^1$ | $9.702 \times 10^1$ | $1.181 \times 10^2$ |
| Python | Top-3 MLP Components (Avg.) | $1.010 \times 10^2$ | $1.311 \times 10^2$ | $1.744 \times 10^2$ |
| | Top-3 Attn Components (Avg.) | $6.978 \times 10^1$ | $9.599 \times 10^1$ | $1.250 \times 10^2$ |
| Science | Top-3 MLP Components (Avg.) | $1.019 \times 10^2$ | $1.343 \times 10^2$ | $1.660 \times 10^2$ |
| | Top-3 Attn Components (Avg.) | $8.013 \times 10^1$ | $9.999 \times 10^1$ | $1.145 \times 10^2$ |

Table 4: Aggregated LoRA weight update norms for the Llama 3.2 1B model across all domains.

| Domain | Component Group | Avg. Full Run Norm | Avg. Ensemble Norm | Top Solo Run Norm |
|---|---|---|---|---|
| CPP | Top-3 MLP Components (Avg.) | $1.131 \times 10^2$ | $1.508 \times 10^2$ | $1.944 \times 10^2$ |
| | Top-3 Attn Components (Avg.) | $8.660 \times 10^1$ | $1.150 \times 10^2$ | $1.389 \times 10^2$ |
| Finance | Top-3 MLP Components (Avg.) | $1.063 \times 10^2$ | $8.506 \times 10^1$ | $1.201 \times 10^2$ |
| | Top-3 Attn Components (Avg.) | $6.711 \times 10^1$ | $5.694 \times 10^1$ | $7.221 \times 10^1$ |
| Math | Top-3 MLP Components (Avg.) | $1.151 \times 10^2$ | $1.607 \times 10^2$ | $2.025 \times 10^2$ |
| | Top-3 Attn Components (Avg.) | $8.405 \times 10^1$ | $1.062 \times 10^2$ | $1.209 \times 10^2$ |
| Medical | Top-3 MLP Components (Avg.) | $1.116 \times 10^2$ | $1.457 \times 10^2$ | $1.798 \times 10^2$ |
| | Top-3 Attn Components (Avg.) | $1.139 \times 10^2$ | $1.196 \times 10^2$ | $1.402 \times 10^2$ |
| Python | Top-3 MLP Components (Avg.) | $1.123 \times 10^2$ | $1.535 \times 10^2$ | $2.042 \times 10^2$ |
| | Top-3 Attn Components (Avg.) | $8.578 \times 10^1$ | $1.173 \times 10^2$ | $1.493 \times 10^2$ |
| Science | Top-3 MLP Components (Avg.) | $1.189 \times 10^2$ | $1.610 \times 10^2$ | $2.009 \times 10^2$ |
| | Top-3 Attn Components (Avg.) | $1.024 \times 10^2$ | $1.278 \times 10^2$ | $1.446 \times 10^2$ |

Table 5: Aggregated LoRA weight update norms for the Gemma-3 4B model across all domains.

| Domain | Component Group | Avg. Full Run Norm | Avg. Ensemble Norm | Top Solo Run Norm |
|---|---|---|---|---|
| CPP | Top-3 MLP Components (Avg.) | $7.481 \times 10^1$ | $8.510 \times 10^1$ | $9.509 \times 10^1$ |
| | Top-3 Attn Components (Avg.) | $4.523 \times 10^1$ | $5.179 \times 10^1$ | $6.092 \times 10^1$ |
| Finance | Top-3 MLP Components (Avg.) | $3.211 \times 10^1$ | $3.883 \times 10^1$ | $4.720 \times 10^1$ |
| | Top-3 Attn Components (Avg.) | $2.398 \times 10^1$ | $2.806 \times 10^1$ | $3.566 \times 10^1$ |
| Math | Top-3 MLP Components (Avg.) | $6.152 \times 10^1$ | $7.033 \times 10^1$ | $7.748 \times 10^1$ |
| | Top-3 Attn Components (Avg.) | $3.345 \times 10^1$ | $3.862 \times 10^1$ | $4.418 \times 10^1$ |
| Medical | Top-3 MLP Components (Avg.) | $7.913 \times 10^1$ | $8.882 \times 10^1$ | $1.060 \times 10^2$ |
| | Top-3 Attn Components (Avg.) | $4.881 \times 10^1$ | $5.361 \times 10^1$ | $6.759 \times 10^1$ |
| Python | Top-3 MLP Components (Avg.) | $7.612 \times 10^1$ | $8.496 \times 10^1$ | $9.706 \times 10^1$ |
| | Top-3 Attn Components (Avg.) | $4.755 \times 10^1$ | $5.305 \times 10^1$ | $6.187 \times 10^1$ |
| Science | Top-3 MLP Components (Avg.) | $8.339 \times 10^1$ | $9.547 \times 10^1$ | $1.049 \times 10^2$ |
| | Top-3 Attn Components (Avg.) | $4.698 \times 10^1$ | $5.223 \times 10^1$ | $5.652 \times 10^1$ |

Table 6: Aggregated LoRA weight update norms for the Gemma-3 1B model across all domains.

| Domain | Component Group | Avg. Full Run Norm | Avg. Ensemble Norm | Top Solo Run Norm |
|---|---|---|---|---|
| CPP | Top-3 MLP Components (Avg.) | $4.315 \times 10^1$ | $5.039 \times 10^1$ | $6.484 \times 10^1$ |
| | Top-3 Attn Components (Avg.) | $2.451 \times 10^1$ | $2.822 \times 10^1$ | $3.337 \times 10^1$ |
| Finance | Top-3 MLP Components (Avg.) | $2.478 \times 10^1$ | $2.891 \times 10^1$ | $3.953 \times 10^1$ |
| | Top-3 Attn Components (Avg.) | $1.691 \times 10^1$ | $1.956 \times 10^1$ | $3.240 \times 10^1$ |
| Math | Top-3 MLP Components (Avg.) | $4.022 \times 10^1$ | $4.570 \times 10^1$ | $5.823 \times 10^1$ |
| | Top-3 Attn Components (Avg.) | $2.003 \times 10^1$ | $2.292 \times 10^1$ | $2.922 \times 10^1$ |
| Medical | Top-3 MLP Components (Avg.) | $4.811 \times 10^1$ | $5.544 \times 10^1$ | $7.106 \times 10^1$ |
| | Top-3 Attn Components (Avg.) | $2.955 \times 10^1$ | $3.401 \times 10^1$ | $4.053 \times 10^1$ |
| Python | Top-3 MLP Components (Avg.) | $4.297 \times 10^1$ | $4.926 \times 10^1$ | $6.502 \times 10^1$ |
| | Top-3 Attn Components (Avg.) | $2.501 \times 10^1$ | $2.846 \times 10^1$ | $3.237 \times 10^1$ |
| Science | Top-3 MLP Components (Avg.) | $4.973 \times 10^1$ | $5.627 \times 10^1$ | $6.923 \times 10^1$ |
| | Top-3 Attn Components (Avg.) | $2.516 \times 10^1$ | $2.830 \times 10^1$ | $3.364 \times 10^1$ |

## A.2 PROBING ANALYSIS

The process of calculating separability scores between each pair of datasets, layer-wise, consists of 2 main components:

1) Hooking to get activations

2) Using these activations to get the Separability Scores

**Hook placement and construction of per-sample representations.** When analyzing representations inside transformer layers, forward hooks are placed on sub-modules corresponding to the **Attention block**, **MLP block**, and **Residual stream activations**. Each hook captures the output tensor of shape $[B, S, D]$, where $B$ is the batch size (examples per forward pass), $S$ is the sequence length (tokens per example), and $D$ is the hidden dimension of the representation. To simplify, the token dimension is mean-pooled, giving a $[B, D]$ embedding for each batch. These embeddings are concatenated across multiple forward passes to construct a design matrix $X \in \mathbb{R}^{N \times D}$, where $N$ is the total number of collected samples. Alongside, a label vector $y \in \{0, \ldots, C-1\}^N$ is created so that each row $X_r$ corresponds to its class label $y_r$.

To compute **Fisher separability** between two classes $i$ and $j$, we first isolate the subsets of $X$ belonging to those labels, giving matrices $X_i \in \mathbb{R}^{n_i \times D}$ and $X_j \in \mathbb{R}^{n_j \times D}$. The mean representation of each class ($\mu_i, \mu_j$) is calculated across their samples, and the variance within each class ($\text{var}_i, \text{var}_j$) is also estimated. Fisher's score is then defined as the squared distance between the two class means, normalized by the sum of their variances. Intuitively, if the means are far apart relative to how spread out the classes are internally, the score is high, indicating that the two classes are well separated in the representation space.

For the **Maximum Mean Discrepancy (MMD)**, the same class-specific subsets $X_i$ and $X_j$ are compared using a kernel function, typically a Gaussian RBF kernel. Pairwise distances between samples are used to determine the kernel bandwidth $\gamma$, and kernel similarity matrices are constructed: within-class ($K_{ii}, K_{jj}$) and cross-class ($K_{ij}$). The MMD score is then computed as the difference between average within-class similarities and average cross-class similarities. A larger MMD value means the two distributions $X_i$ and $X_j$ are more dissimilar, capturing not just differences in means but also higher-order mismatches in distributional shape.

### EXPERIMENT PARAMETERS

| Samples per domain (forward pass) | MLP hook | Attention hook | Batch size |
|:---:|:---:|:---:|:---:|
| 1000 | $up\_proj$ | $o\_proj$ | 8 |

Parameters used for all models: Llama 3.2 3B, Llama 3.2 1B, Gemma 3 4B, and Gemma 3 1B.

## B DATASETS

**C++, Python**  For our coding datasets, we have used the Open Coder LLM Annealing Corpus (Huang et al. (2024)) which contains functional code snippets on various coding questions. This dataset aligns with our Human Benchmark Evaluation tests since it uses the same formatting. Each data point has a top level comment describing the task followed by a function that implements the task. The original dataset also contains inline comments inside the function body but these have been striped for conciseness. Listing 1 and Listing 10 showcase examples from our dataset on C++ and Python snippets.

**Science**  We have used the SciQ dataset (Johannes Welbl, 2017) which contains crowd-sourced questions on Physics, Chemistry and Biology. The questions are in multiple-choice format with 4 answer options each. For our purposes we have formatted the data-points into Context, Question and Answer.

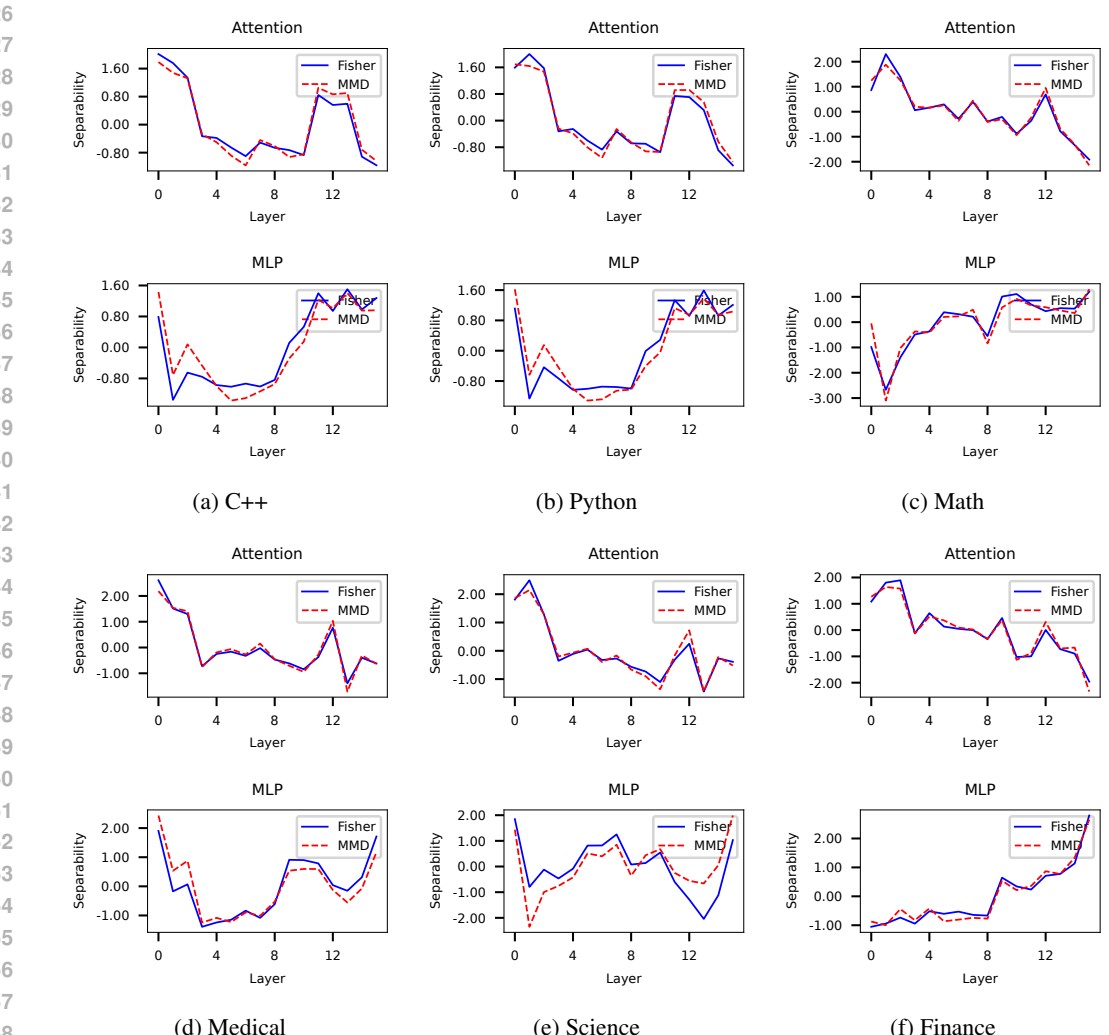

Figure 7: probe separability results for Llama 1B Model

```
Context: Enzymes are critical to the body's healthy functioning.
    They assist, for example, with the breakdown of food and its
    conversion to energy. In fact, most of the chemical reactions
    in the body are facilitated by enzymes.
Question: Most of the chemical reactions in the body
    are facilitated by what?
Options: A. proteins B. enzymes C. vitamins D. carbohydrates
Answer: B
```

**Mathematics**  The Math dataset is GSM8K (Cobbe et al., 2021a) which is a dataset of 8.5k high quality math word problems. The dataset contains question answering on basic mathematical problems that require multi-step reasoning. The datapoints are also similarly formatted into Question, Answer and Final Answer.

```
Question: Natalia sold clips to 48 of her friends in April, and
    then she sold half as many clips in May. How many clips did
    Natalia sell altogether in April and May?
Answer: Natalia sold 48/2 = <<48/2=24>>24 clips in May. Natalia
    sold 48+24 = <<48+24=72>>72 clips altogether in April and May. #### 72
Final Answer: 72.
```

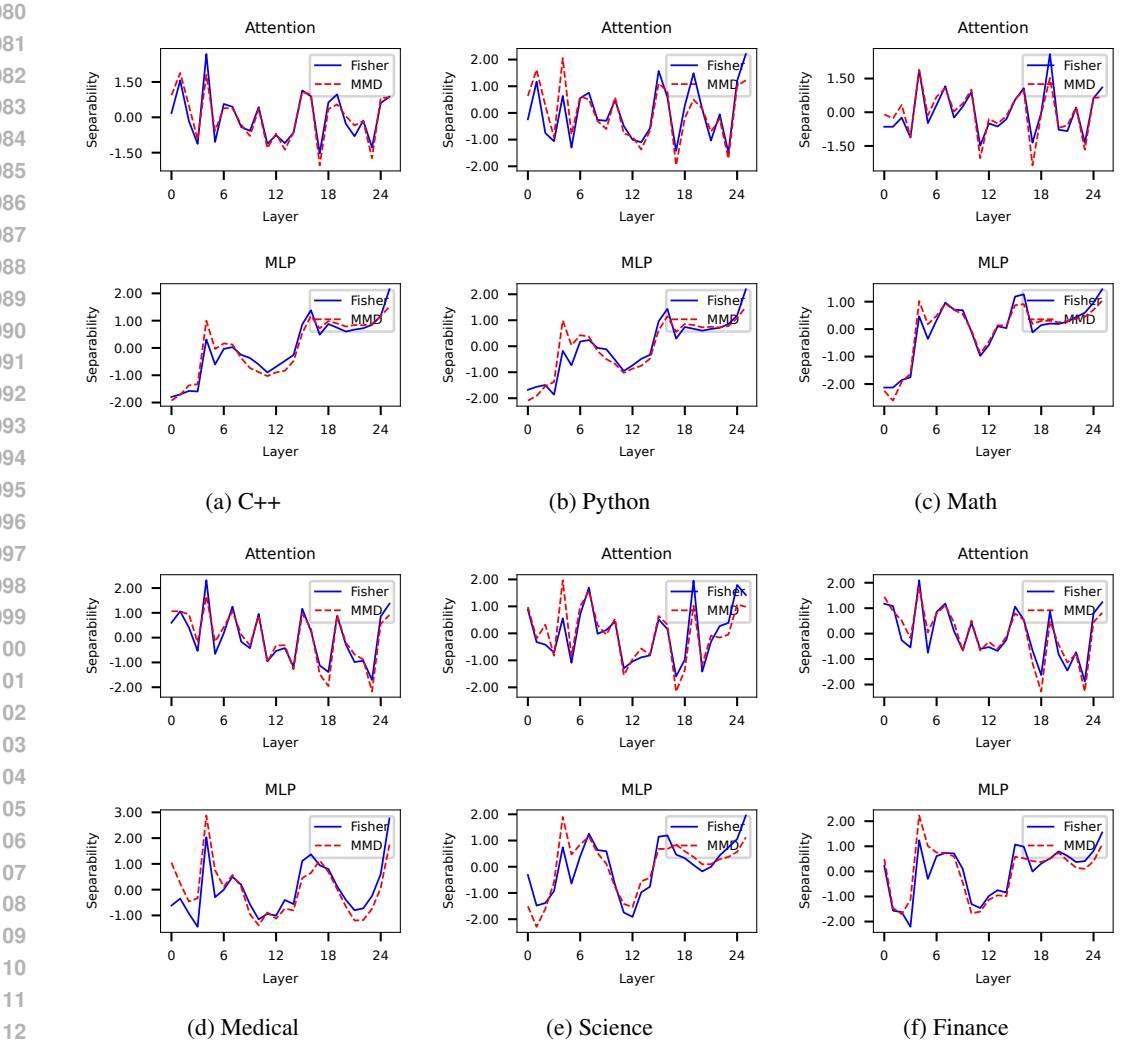

Figure 8: probe separability results for Gemma 1B Model

**Finance**    The Finance dataset (Mateega et al., 2025) is a set of financial question and answer pairs extracted from company annual reports, balance sheets, and financial statements.The datapoints contain context with some financial values and the model is questioned upon some value that is dependant on this information. A similar formatting technique is used where we explicitly state the context, question and answer.

```
Context: Liabilities: 8,537.39 Total Capital And Liabilities:
    13,410.53 ASSETS: nan NON-CURRENT ASSETS: nan Tangible
    Assets: 74.2 Intangible Assets: 4.16 Capital Work-In-Progress:
    0 Other Assets: 0 Fixed Assets: 98.73 Non-Current
    Investments: 0 Deferred Tax Assets [Net]: 0 Long Term Loans And Advances: 0
    Other Non-Current Assets: 15.61 Total Non-Current Assets: nan
Question: What is the total value of assets of the company?
Answer: The total value of assets of the company is $13,410.53.
Final Answer: 13410.53.
```

**Medical**    We use the ReasonMed dataset (link lingshu-medical-mllm/ReasonMed) which is an open-source synthetic medical reasoning dataset containing multi-step chain-of-thought (CoT) rationales and concise summaries of LLMs such as Qwen-2.5-72B, DeepSeek-R1-Distill-Llama-70B, and HuatuoGPT-o1-70B on medical questions.

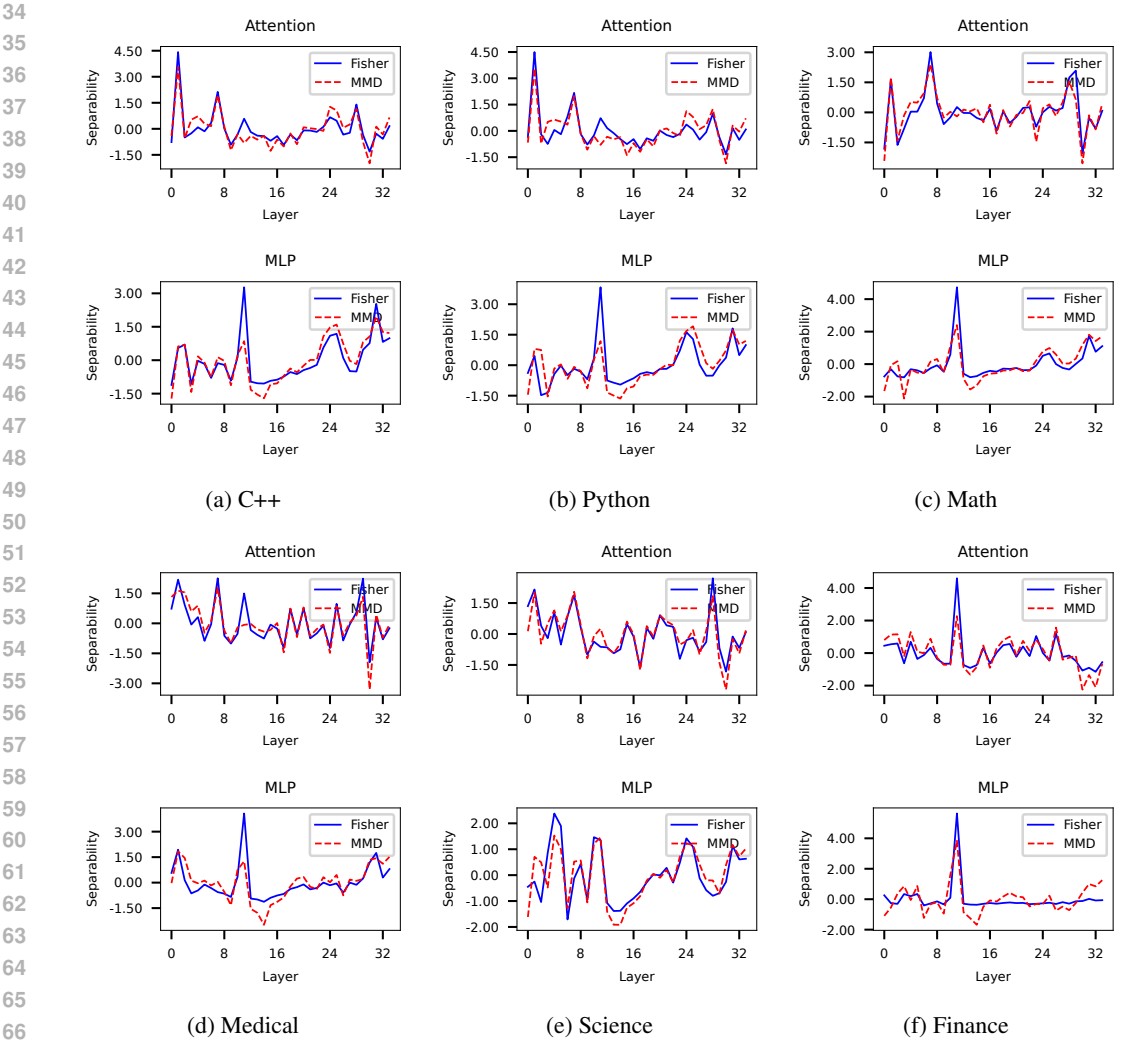

Figure 9: probe separability results for Gemma 4B Model

```
The question presents a radiographic scenario: a PA
(posteroanterior) ulnar deviation view of the wrist, asking
for the most likely diagnosis among the following options:
Osteomyelitis, De Quervain tenosynovitis, Hypertrophic
osteoarthropathy, and Rheumatoid arthritis. The correct answer
is De Quervain tenosynovitis. To comprehensively understand
and justify this answer, it's essential to dissect
each component... (truncated)
```

## C  EVALUATION

### C.1  EVALUATION RESULTS

Our evaluation of domain-specific performance uses two accuracy metrics tailored to the task type. For the Math, Science, Finance, and Medical domains, we report standard classification accuracy, defined as:

$$\text{Accuracy} = \frac{\text{Number of Correct Predictions}}{\text{Total Number of Samples}}$$

For the programming domains (C++ and Python), we evaluate code generation correctness using the **pass@k** metric. Specifically, we use **pass@10**, where the model generates 10 candidate solutions for

```python
def find_pivot_index(nums: list[int]) -> int:
        Finds the pivot index of a list of numbers.

    The pivot index is where the sum of the numbers to the left of
     the index
    is equal to the sum of the numbers to the right of the index.

    Args:
        nums: A list of integers.

    Returns:
        The pivot index if one exists, otherwise -1.

    Examples:
        >>> find_pivot_index([1, 7, 3, 6, 5, 6])
        3
        >>> find_pivot_index([1, 2, 3])
        -1
        >>> find_pivot_index([2, 1, -1])
        0

    total_sum = sum(nums)
    left_sum = 0

    for i, num in enumerate(nums):
        if left_sum == total_sum - left_sum - num:
            return i
        left_sum += num

    return -1
```

Listing 1: A Python snippet from our dataset illustrating a simple coding problem with a doc-string that explicitly describes the working of the function

```cpp
// This function takes a positive integer as input and returns a
    list of its prime factors,
// which are the prime numbers that multiply together to equal the
    original number.
// The prime factors are returned in ascending order.
//
// Parameters:
// * number: A positive integer to be factorized into its prime
    factors.
//
// Examples:
// * find_prime_factors(8) => [2, 2, 2]
// * find_prime_factors(25) => [5, 5]
// * find_prime_factors(70) => [2, 5, 7]
std::vector<int> find_prime_factors(int number) {
    std::vector<int> prime_factors;

    while (number % 2 == 0) {
        prime_factors.push_back(2);
        number /= 2;
    }

    for (int i = 3; i <= std::sqrt(number) + 1; i += 2) {
        while (number % i == 0) {
            prime_factors.push_back(i);
            number /= i;
        }
    }

    if (number > 2) {
        prime_factors.push_back(number);
    }

    return prime_factors;
}
```

Figure 10: A C++ snippet from our dataset featuring a prime factorization problem. Each example contains a descriptive comment above the function body and clear naming conventions for the function itself.

each problem. A problem is considered solved if at least one of these candidates passes all unit tests. The accuracy is therefore calculated as:

$$\text{pass@10} = \frac{\text{Number of Problems with at least one passing solution}}{\text{Total Number of Problems}}$$

It is important to note that the results presented, particularly for the smaller 1B models, may exhibit some noise. These models operate with fewer parameters, making performance sensitive to minor variations in fine-tuning, which can affect the robustness of the generated outputs.

|         | PT    | MLP   | Attn  | Both  | Top-1 MLP | Top-1 Attn | Top-3 MLP | Top-3 Attn |
|---------|-------|-------|-------|-------|-----------|------------|-----------|------------|
| Math    | 0.040 | **0.070** | 0.050 | 0.020 | 0.050     | 0.040      | 0.030     | 0.040      |
| Science | 0.395 | 0.390 | **0.535** | 0.475 | 0.385     | 0.290      | 0.310     | 0.325      |
| CPP     | 0.120 | 0.020 | 0.020 | 0.000 | 0.050     | 0.040      | **0.130** | 0.040      |
| Python  | 0.440 | 0.020 | 0.180 | 0.040 | 0.040     | **0.350**  | 0.160     | 0.290      |
| Finance | 0.180 | 0.020 | 0.010 | 0.000 | 0.060     | 0.040      | 0.020     | **0.070**  |
| Medical | 0.847 | 0.687 | 0.787 | 0.813 | 0.904     | 0.424      | **0.916** | 0.864      |

**Llama-3.2-1B**

|         | PT    | MLP   | Attn  | Both  | Top-1 MLP | Top-1 Attn | Top-3 MLP | Top-3 Attn |
|---------|-------|-------|-------|-------|-----------|------------|-----------|------------|
| Math    | 0.100 | 0.030 | 0.060 | 0.030 | **0.140** | 0.060      | 0.040     | 0.040      |
| Science | 0.625 | **0.755** | 0.700 | 0.650 | 0.610     | 0.600      | 0.425     | 0.425      |
| CPP     | 0.320 | 0.000 | 0.030 | 0.000 | 0.000     | **0.286**  | 0.000     | 0.150      |
| Python  | 0.470 | 0.040 | 0.300 | 0.050 | 0.286     | **0.371**  | 0.220     | 0.340      |
| Finance | 0.080 | 0.020 | **0.050** | 0.040 | 0.025     | 0.000      | 0.030     | 0.030      |
| Medical | 0.900 | 0.713 | **0.912** | 0.512 | 0.880     | 0.880      | **0.912** | 0.888      |

**Gemma-3-1B**

|         | PT    | MLP   | Attn  | Both  | Top-1 MLP | Top-1 Attn | Top-3 MLP | Top-3 Attn |
|---------|-------|-------|-------|-------|-----------|------------|-----------|------------|
| Math    | 0.080 | 0.080 | 0.060 | 0.080 | 0.200     | 0.133      | **0.400** | 0.267      |
| Science | 0.715 | 0.780 | 0.780 | 0.760 | **0.840** | 0.820      | 0.760     | 0.700      |
| CPP     | 0.833 | 0.033 | 0.000 | 0.000 | 0.028     | **0.457**  | 0.000     | 0.286      |
| Python  | 0.300 | 0.233 | 0.333 | 0.033 | **0.371** | 0.343      | 0.286     | 0.343      |
| Finance | 0.040 | 0.000 | 0.000 | 0.000 | 0.000     | 0.000      | **0.025** | **0.025**  |
| Medical | 0.925 | **0.950** | **0.950** | 0.300 | 0.875     | **0.950**  | 0.725     | 0.850      |

**Gemma-3-4B**

As an alternative performance metric, we measured the asymptotic validation loss for different component combinations. The results aligned with our separability analysis: layers identified as having high activation separability consistently outperformed those with lower separability, converging to a significantly lower validation loss.

## C.2 DOMAIN EVALUATION

### C.2.1 MATH

**Dataset chosen:** *GSM8K* (Grade School Math 8K) introduced by Cobbe et al. (2021b) is a collection of grade-school level math word problems designed to evaluate multi-step arithmetic and reasoning ability. The dataset emphasizes chain-of-thought style reasoning where intermediate steps are useful to arrive at the correct numeric result.

GSM8K is used here as it's a widely used benchmark for studying reasoning behavior in language models and for evaluating self-consistency / majority-vote sampling methods. Also, it is not too difficult, hence used for evaluation on the small models considered.

**Prompt–Output Illustration:**

```
# <prefix text (8-shot demos)
    provided for context>

Q: John has 3 apples.
He buys 2 more.
How many apples does he have
    now?

A: Let's reason step by step.
At the end, give the final
    numeric
answer on its own line in
    this exact format:
#### <number>
Answer:
```

```
# Example reasoning and
    output

Step 1: John starts with 3
    apples.
Step 2: He buys 2 more.
Step 3: Total apples = 3 + 2
    = 5.

#### 5
Answer:
```

Illustration of the prompt (left) and an example of the expected LLM output (right).

| Evaluation Samples | Sampling Amount Per Sample | Max Generation Tokens | Temperature | Top_p |
|---|---|---|---|---|
| 100 | 10 | 1024 | 0.7 | 0.90 |

(Hyper-Parameters used during Model inference For Evaluation(Self-Consistency)

### C.2.2 FINANCE

**Dataset chosen:** *FinanceQA* introduced by Mateega et al. (2025) is a curated set of financial question–answer pairs extracted from company filings (annual reports, balance sheets, and reports). It supplies queries, short factual answers, and the supporting context passage from the source document (e.g., a few sentences or table rows). Focus is on **numerical output comparison and extraction**.

FinanceQA is used for evaluation as it provides a domain-specific **"finance + math"** evaluation setting, requiring both factual retrieval and quantitative reasoning.

**Prompt–Output Illustration:**

```
# FinanceQA prompt builder

(context + query)

Context:
<supporting passage from
    financial filings>

Question:
<query here>

Answer: The final answer is

Final Answer:
```

```
# Example reasoning and
    output

Step 1: From the context, the
    net profit
margin in 2021 is explicitly
    given.
Step 2: The reported margin
    is 11.04%.

Final Answer: 11.04%
```

Illustration of the FinanceQA prompt template (left) and an example expected LLM output (right).

| Evaluation Samples | Sampling Amount Per Sample | Max Generation Tokens | Temperature | Top_p |
|---|---|---|---|---|
| 100 | 10 | 512 | 0.7 | 0.95 |

(Hyper-Parameters used during Model inference For Evaluation(Self-Consistency))

### C.2.3 MEDICAL

**Dataset chosen:** *PubMedQA* introduced by Jin et al. (2019) is a dataset of biomedical research questions paired with contexts and a short (yes/no) final decision derived from biomedical articles. Each sample often contains an abstract or supporting passage and a question about the clinical finding; the ground truth is typically a binary decision. Sometimes if LLM is highly undecisive the output of LLM is assumed 'None'

We use PubMedQA because it is a widely-used , biomedical QA benchmark for evaluating concise, high-precision yes/no answers in the clinical/research domain.

**Prompt–Output Illustration:**

```
# PubMedQA prompt builder (
    question + context)

Context:
<concatenated context sentences
    or abstract>

Question: <question here>

Based on the context above,
    answer the question
with exactly 'yes' or 'no' (
    lowercase),
and do NOT provide any
    explanation.
Answer:
```

Illustration prompt template used Sample output is simply Yes/No , In case Bad output Then None is interpreted

| Evaluation Samples | Sampling Amount Per Sample | Max Generation Tokens | Temperature | Top_p |
|---|---|---|---|---|
| 250 | 1 | 512 | 0.0 | 1.00 |

(Hyper-Parameters used during Model inference For Evaluation (Greedy))

### C.2.4 SCIENCE

**Dataset chosen:** *SciQ* introduced by Welbl et al. (2017) is a data set of multiple choice science questions that contains short grade-level science questions with four answer options (A–D) and optional supporting facts. Each example includes a question, four candidate answers, and (sometimes) a support passage.

SciQ is used because it provides well-formed multiple-choice prompts suitable for evaluation,it is easy for a small LLM hence it is used.

**Prompt–Output Illustration:**

```
// SciQ prompt builder (
    question + options)

Question:
<question text>

Options:
A. <option A>
B. <option B>
C. <option C>
D. <option D>

Answer with the letter of the
    correct option only (A, B,
    C, or D).
Do NOT provide any explanation.
Answer:
```

```
Answer:B
```

Illustration: left = prompt template used for SciQ , model output is a single letter A/B/C/D.

| Evaluation Samples | Sampling Amount Per Sample | Max Generation Tokens | Temperature | Top_p |
|---|---|---|---|---|
| 200 | 1 | 256 | 0.0 | 1.00 |

(Hyper-Parameters used during Model inference For Evaluation(Greedy))

### C.2.5 PYTHON

**Dataset chosen:** *HumanEvalPack (multilingual / Python subset)* Introduced by Chen et al. (2021) is a collection of programming problems with formal problem descriptions, expected function signatures, and test harnesses.

Inputs in the form of coding questions are provided, and the model is expected to output corresponding code which is executed against test cases. The accuracy used for evaluation is **pass@k**, a standard metric for code-generation tasks, rather than simple string-matching accuracy.

HumanEvalPack is used here because it provides language-specific (C++/Python/etc.) prompts with a standard "declaration + examples + tests" scheme. The problems are relatively simple, making this dataset ideal for comparing small models on code generation and correctness.

**Prompt–Output Illustration:**

```
# Problem:
<prompt_or_instruction>

# Signature:
<signature>

# Docstring:
<docstring>

# Examples:
<example_test>

Write the complete Python
    function
implementation only.
Output only valid Python code
    for the
function (no explanation, no
    tests,
no surrounding markdown).
Make sure the function name and
signature match the signature
    above.

Implementation:
```

```
# Example implementation for:
# def add(a: int, b: int) ->
    int

def add(a: int, b: int) -> int:
    # simple implementation
    return a + b
```

Illustration of the Python prompt template (left) and an example expected LLM output (right).

| Evaluation Samples | Sampling Amount Per Sample | Max Generation Tokens | Temperature | Top_p |
|---|---|---|---|---|
| 100 | 10 | 1024 | 0.7 | 0.95 |

(Hyper-Parameters used during Model inference For Evaluation(Self-Consistency))

### C.2.6 CPP

**Dataset chosen:** *HumanEvalPack (multilingual / C++ subset)* Introduced by Chen et al. (2021) is a collection of programming problems with formal problem descriptions, expected function declarations/signatures, and test harnesses .Inputs in the form of coding questions are provided, and the model is expected to output corresponding code which is compiled against test cases.

HumanEvalPack is used here because it provides language-specific (C++/Python/etc.) prompts with a standard "declaration + examples + tests" scheme. The problems are relatively simple, making this dataset ideal for comparing small models on code generation and correctness.

**Prompt–Output Illustration:**

```
// Problem:
<prompt_or_instruction>

// Declaration:
<declaration>

// Docstring / Notes:
<docstring>

// Examples:
<example_test>

Write the C++ implementation
    only
(no explanation, no tests, no
    surrounding markdown).
Include necessary #include
    lines if needed.
Ensure function name and
    signature match the
    declaration above.

Implementation:
```

```
#include <bits/stdc++.h>
using namespace std;

// Example implementation for:
    int add(int a, int b)
int add(int a, int b) {
    // simple implementation
    return a + b;
}
```

Illustration of the C++ prompt template (left) and an example expected LLM output (right).

| Evaluation Samples | Sampling Amount Per Sample | Max Generation Tokens | Temperature | Top_p |
|---|---|---|---|---|
| 100 | 10 | 1024 | 0.7 | 0.95 |

(Hyper-Parameters used during Model inference For Evaluation(Self-Consistency))

# D  EXPERIMENTAL SETUP

## D.1  FINE TUNING

All experiments were run on NVIDIA H100 GPUs, using PyTorch and the Hugging Face 'transformers' and 'peft' libraries. To maximize computational throughput, the model was JIT-compiled using 'torch.compile()'. A fixed set of hyperparameters, detailed in Table 7, was used across all experiments to ensure fair comparison.

Table 7: Common hyperparameters for all fine-tuning experiments.

| Parameter | Value |
|---|---|
| **Training Configuration** | |
| Optimizer | AdamW |
| Learning Rate | $1 \times 10^{-3}$ |
| Batch Size | 8 |
| Epochs (Stage 1 Mapping) | 10 |
| Epochs (Stage 2 Validation) | 3 |
| Seed | 42 |
| Precision | 'bfloat16' |
| **LoRA Configuration** | |
| Rank ($r$) | 16 |
| Alpha ($\alpha$) | 32 ($2 \times r$) |
| Dropout | 0.05 |
| Target Modules (Attn) | `q_proj, k_proj, v_proj, o_proj` |
| Target Modules (MLP) | `gate_proj, up_proj, down_proj` |

## D.2 PROBING ANALYSIS

In addition to Fisher Separability and Maximum Mean Discrepancy (MMD), we also evaluated probing separability using other metrics such as classification probing accuracy, cosine similarity, and V-bits. However, for high-level abstraction tasks such as *Domain Separability*, the results across layers were not clearly distinguishable. This arises because, in such tasks, the points in the activation hyperspace are widely dispersed. Consequently, strong metrics such as V-bits or probing classification accuracy can easily separate these spread-out representations, making them less informative for fine-grained layer-wise analysis. In contrast, weaker metrics such as Fisher separability and MMD are more useful in these cases, as they provide more sensitive distinctions when the data is already well separated.

On the other hand, for low-level abstraction tasks such as *Concept-level Separability*, the points in the activation hyperspace are closely packed. In these scenarios, strong metrics such as V-bits prove more effective, yielding clearly distinguishable results across layers. This observation is consistent with findings reported in Ju et al. (2024b).

# E    EXTENDED DISCUSSION

## E.1    HYDRA EFFECT

The Hydra Effect describes a form of self-repair capability present in LLMs. As described by McGrath et al. (2023), it refers to the mismatch between a layer's apparent contribution (measured by projecting its activations through the unembedding mechanism, $\Delta_{unembed}$) and its functional importance (measured by ablating the layer, $\Delta_{ablate}$). We expect the ablation to reduce the model's confidence proportionally to its apparent contribution, but downstream layers reconstruct the corrupted signal so that

$$\Delta_{\text{ablate,l}} < \Delta_{\text{unembed,l}}$$

During interventions, the KL divergence is lower for early layers with high fisher score due to this reason since the intervention done is reverted to some extent by downstream layers.

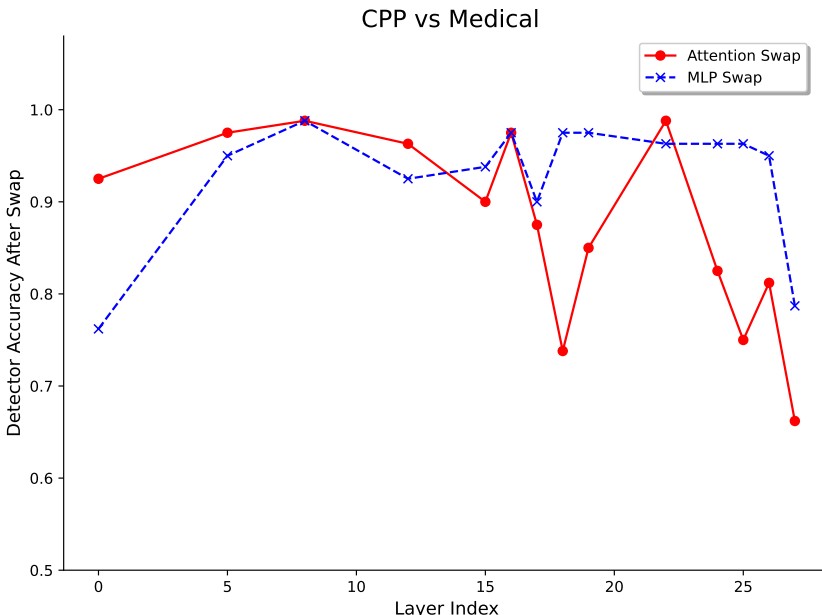

Figure 11: **Causal Impact of Attention and MLP Swaps on Domain Classification.** The plot shows the results of our last-token Causal Signal Probe for the CPP vs. Medical domains. Accuracy reflects the model's ability to correctly classify the original text after a single-token activation swap. A lower accuracy indicates a more effective causal intervention (i.e., the swap successfully steered the model's output).

To directly test the causal role of different components and investigate this self-repair phenomenon, we performed a Causal Signal Probe. As shown in Figure 11 for the CPP vs. Medical pair, we found that swapping the last-token activations from the Attention stream had a dramatically larger impact on the final classification than swapping MLP activations, particularly in the latter half of the network. While interventions in early layers were largely ignored by the model (accuracy ≈ 1.0), consistent with the Hydra effect, swapping Attention activations in layers 15-20 caused a significant drop in accuracy. This demonstrates that these late-stage attention blocks possess a powerful causal steering handle over the model's final domain representation, whereas the influence of individual MLP blocks is less critical.

## E.2    METHODOLOGY FOR THE LAST-TOKEN CAUSAL SIGNAL PROBE

### E.2.1    OBJECTIVE

The Causal Signal Probe was designed to move beyond correlational measures and establish a direct causal link between the activations of specific model components and the model's final domain classification. The primary goal was to answer the question: "If we forcibly inject information from Domain B into the processing of a text from Domain A at a specific layer L, does the model's final thought flip to Domain B?" This experiment allows us to identify which layers and components act as influential steering handles for domain representation.

### E.2.2    EXPERIMENTAL DESIGN

The probe consists of three main stages: (1) training a reliable Truth Detector to classify the model's final hidden state, (2) performing a targeted causal intervention via activation swapping, and (3) measuring the effect of this intervention using the Truth Detector.

**Stage 1: Training the Truth Detector**    To create an objective arbiter of the model's final representation, we first trained a simple linear probe, which we term the Truth Detector.

- **Purpose:** The detector's sole job is to look at the final hidden-state vector of the model and classify which of the two domains it belongs to.

- **Data Generation:** We passed 500 text samples from the `cpp` domain and 500 from the `medical` domain through the frozen Llama-3.2-3B model. We collected the final hidden-state activation vector for each sample.

- **Architecture:** The detector is a simple linear model that maps to 2 output logits (one for each domain).

- **Training:** The collected activations were split into an 80/20 train/validation set. The detector was trained for 7 epochs using an Adam optimizer and Cross-Entropy loss. For the CPP vs. Medical pair, the detector rapidly achieved 100.0% validation accuracy, confirming it as a highly reliable ground truth for our experiment.

**Stage 2: The Causal Intervention (Last-Token Activation Swap)**   This is the core of the causal experiment. We performed a precise activation swap for each layer and component under investigation.

- **Procedure:** For each trial, a `base_text` (true label) and a `donor_text` (opposing domain) are randomly selected.

- The `donor_text` is passed through the model up to a target layer $L$. We cache the activation vector of its very last token for a specific component (e.g., the output of the attention block).

- The `base_text` is then passed through the model. Using PyTorch forward hooks, we intercept the computation at layer $L$, right after the target component (Attention or MLP) has finished its computation.

- The hook replaces the last-token activation vector of the `base_text` with the cached vector from the `donor_text`. All other token activations remain unchanged.

- The forward pass resumes, processing this patched sequence representation, and the final hidden-state vector is collected.

This last-token methodology is crucial as it is a clean, minimal intervention that directly targets the same vector representation our Truth Detector was trained on, avoiding confounding issues related to variable sequence lengths.

### E.2.3    DETAILED INTERPRETATION OF THE CPP VS. MEDICAL GRAPH

The provided graph (Figure 11) plots the Detector Accuracy After Swap for the CPP vs. Medical domain pair.

- **Overall Trend:** The most striking feature is the growing divergence between the Attention and MLP swaps. While both start with high accuracy, the Attention swap becomes significantly more impactful (lower accuracy) in the later layers.

- **Early Layers (0-8):** Interventions here have minimal effect (Accuracy 76%-99%), indicating that the initial distributed domain signal overpowers the single-token swap, and downstream layers repair the representation.

- **Mid-Layers (8-18):** A crucial divergence begins. MLP swap impact remains low, while Attention swap accuracy starts to dip, suggesting attention mechanisms here begin to refine the domain-specific representation.

- **Late Layers (18-27):** This region shows the strongest causal effect. The Attention swap becomes highly volatile and effective. This confirms that some attention layers act as powerful steering handles, whereas the MLP influence is secondary and less decisive.

### E.3    CHARACTERISTIC TOKENS

The process of selecting characteristic tokens is derived from the same causal intervention process done in reverse. Instead of finding layers that do the most change to specific tokens, we find tokens that are most sensitive to interventions on all layers. This process is coined as the reverse causal intervention on a model.

When we do an intervention on a single layer from one domain to another, the tokens of the new domain are shifted up in probability. The overall shift across the vocabulary is averaged across all layers and the Top-k "promoted" tokens are saved in a list for the intervening dataset. For example, we have found when intervening C++ prompts with Python activations, tokens such as $def$, $import$ and $python$ are promoted. These form the characteristic token set for Python and this set is used in our causal intervention experiments further on.

### E.4 DELTA BIAS

Let $V$ be the entire vocabulary of the model. We denote the probability associated with a subset of vocabulary $S \subset V$ as $P(S|x) = \sum_{i \in S} p(i|x)$ with a prompt $x$. Suppose we perform the intervention $x_A \overset{l}{\leftarrow} x_B$ where activations of prompt of domain B are inserted into the forward pass of A at layer $l$. Before intervention, $P_{base}(S_A|x_a)$ and $P_{base}(S_B|x_a)$ denote the probabilities of characteristic tokens of A and B before intervention, and $P_{swap}(S_A|x_A \overset{l}{\leftarrow} x_B)$ and $P_{swap}(S_B|x_A \overset{l}{\leftarrow} x_B)$ as the probabilities of the set of characteristic tokens of A and B after intervention. The $Bias$ present in the probability distribution is defined as Bias $= P(S_B) - P(S_A)$. This represents the model's preference on predicting the intervening subset of tokens.

$$\text{Bias}_{base}(x_A) = P_{base}(S_B|x_A) - P_{base}(S_A|x_A)$$

$$\text{Bias}_{swap}(x_A \overset{l}{\leftarrow} x_B) = P_{swap}(S_B|x_A \overset{l}{\leftarrow} x_B) - P_{swap}(S_A|x_A \overset{l}{\leftarrow} x_B)$$

$$\Delta\text{Bias}(A \overset{l}{\leftarrow} B) = \mathbb{E}_{x_A \sim A, x_B \sim B}\left[\text{Bias}_{swap}(x_A \overset{l}{\leftarrow} x_B) - \text{Bias}_{base}(x_A)\right]$$

In our results, we use the convention for when $A \overset{l}{\leftarrow} B$ is done, we plot bias with a positive sign, and when we do intervention $B \overset{l}{\leftarrow} A$, we plot bias with a negative sign to preserve perspective with respect to the set of characteristic tokens B. So, all bias computations are visualized as the shift in preference of B over A.

### E.5 CAUSAL INTERVENTION VARIATIONS

We extend our causal intervention study to other domain pairs beyond the C++/Python case discussed in the main text. Below we present comprehensive results for all tested domain pairs across all four models: Llama-3.2-3B, Llama-3.2-1B, Gemma-3-4B, and Gemma-3-1B.

### E.5.1 LLAMA-3.2-3B: ALL DOMAIN PAIRS

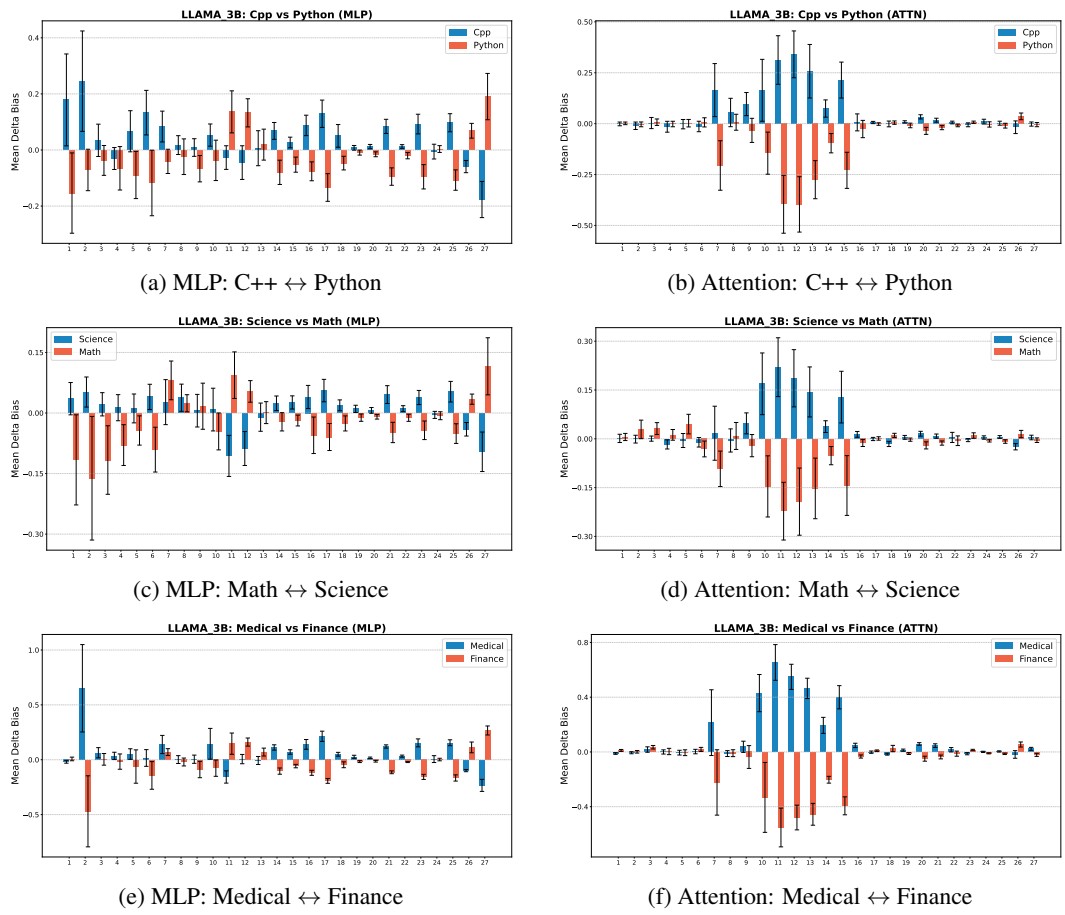

(a) MLP: C++ ↔ Python

(b) Attention: C++ ↔ Python

(c) MLP: Math ↔ Science

(d) Attention: Math ↔ Science

(e) MLP: Medical ↔ Finance

(f) Attention: Medical ↔ Finance

Figure 12: **Llama-3.2-3B causal intervention results across all domain pairs.** Each row shows a different domain pair. Left column: MLP activations exhibit flat, high-variance profiles centered near zero (disruption without directional control). Right column: Attention activations show sharp, localized peaks at mid-depth layers (e.g., 13-15, 23-25), indicating sparse routing hotspots. The pattern is consistent across all three domain pairs, supporting domain-agnostic routing mechanisms.

### E.5.2 LLAMA-3.2-1B: ALL DOMAIN PAIRS

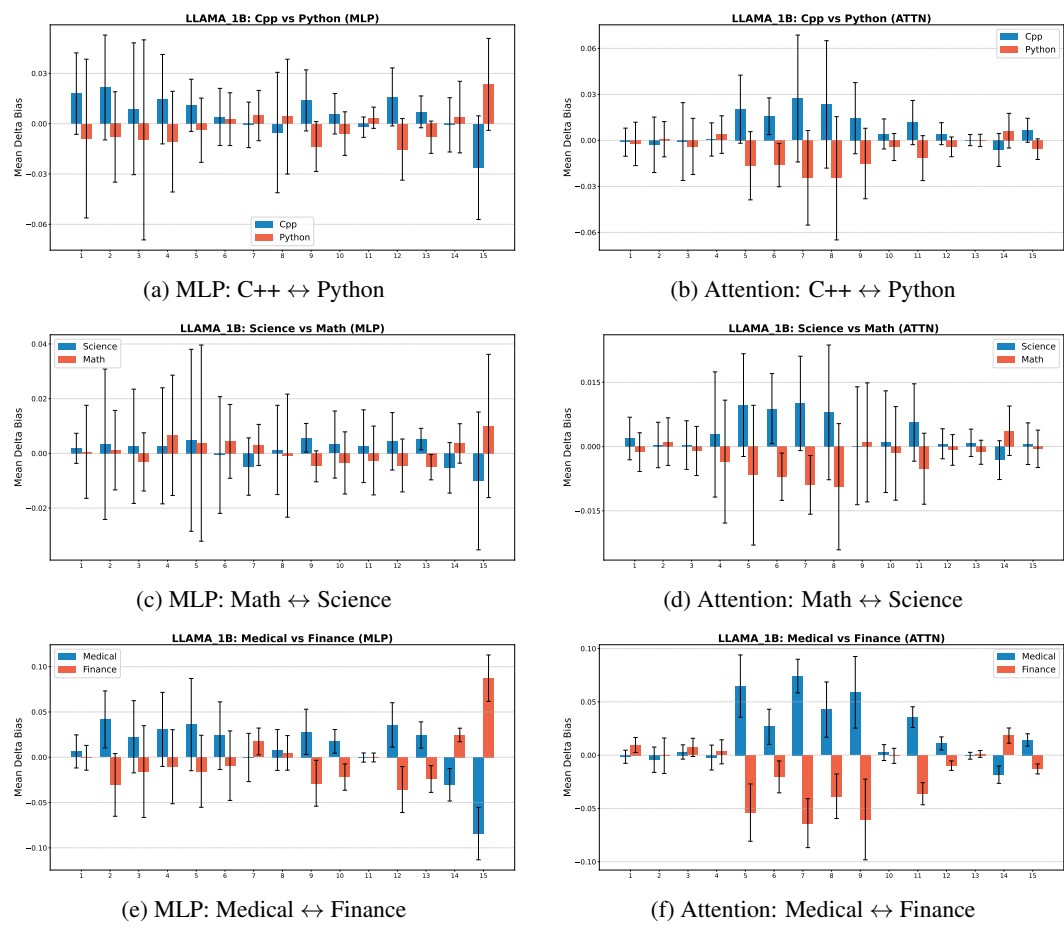

(a) MLP: C++ ↔ Python

(b) Attention: C++ ↔ Python

(c) MLP: Math ↔ Science

(d) Attention: Math ↔ Science

(e) MLP: Medical ↔ Finance

(f) Attention: Medical ↔ Finance

Figure 13: **Llama-3.2-1B causal intervention results across all domain pairs.** Despite the smaller model size (1B parameters, 16 layers), the same qualitative pattern emerges: MLP swaps produce non-directional disruption, while attention swaps yield localized steering peaks. The reduced depth results in fewer but proportionally similar routing layers.

### E.5.3 GEMMA-3-4B: ALL DOMAIN PAIRS

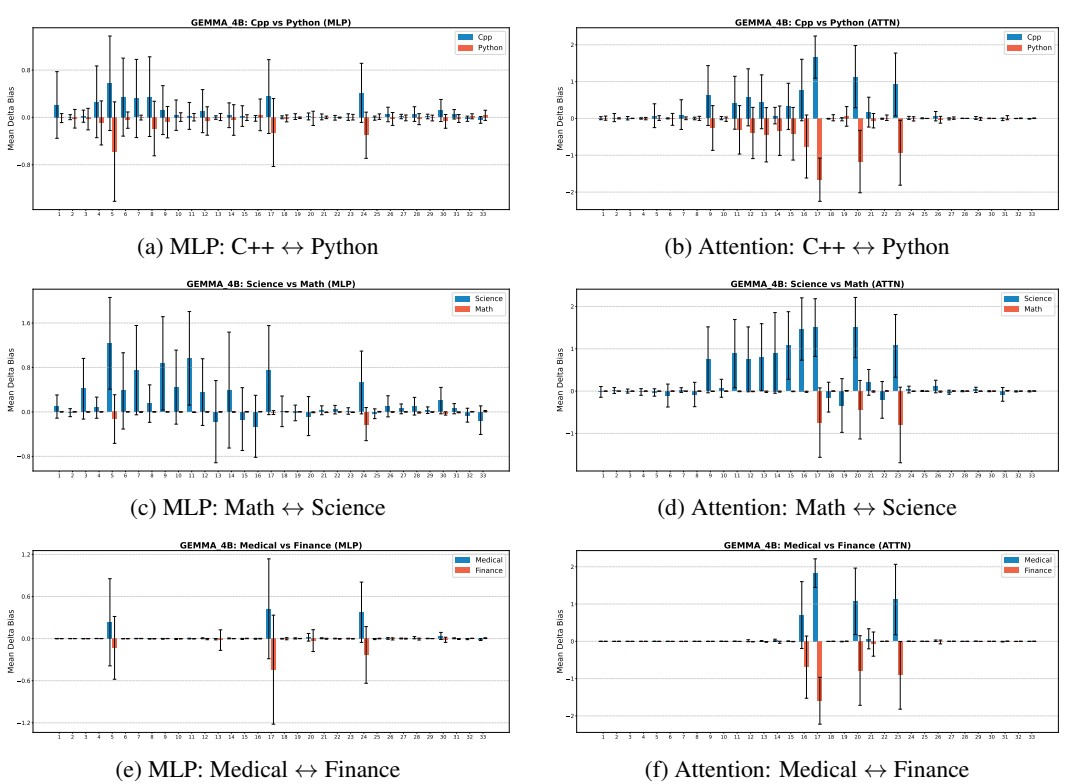

(a) MLP: C++ ↔ Python

(b) Attention: C++ ↔ Python

(c) MLP: Math ↔ Science

(d) Attention: Math ↔ Science

(e) MLP: Medical ↔ Finance

(f) Attention: Medical ↔ Finance

Figure 14: **Gemma-3-4B causal intervention results across all domain pairs.** Gemma models exhibit sharper, more concentrated attention peaks compared to Llama models, suggesting a more specialized hub-like routing architecture. This acute localization is particularly visible in the Medical/Finance pair (bottom right), where a single attention layer dominates the steering signal.

### E.5.4 GEMMA-3-1B: ALL DOMAIN PAIRS

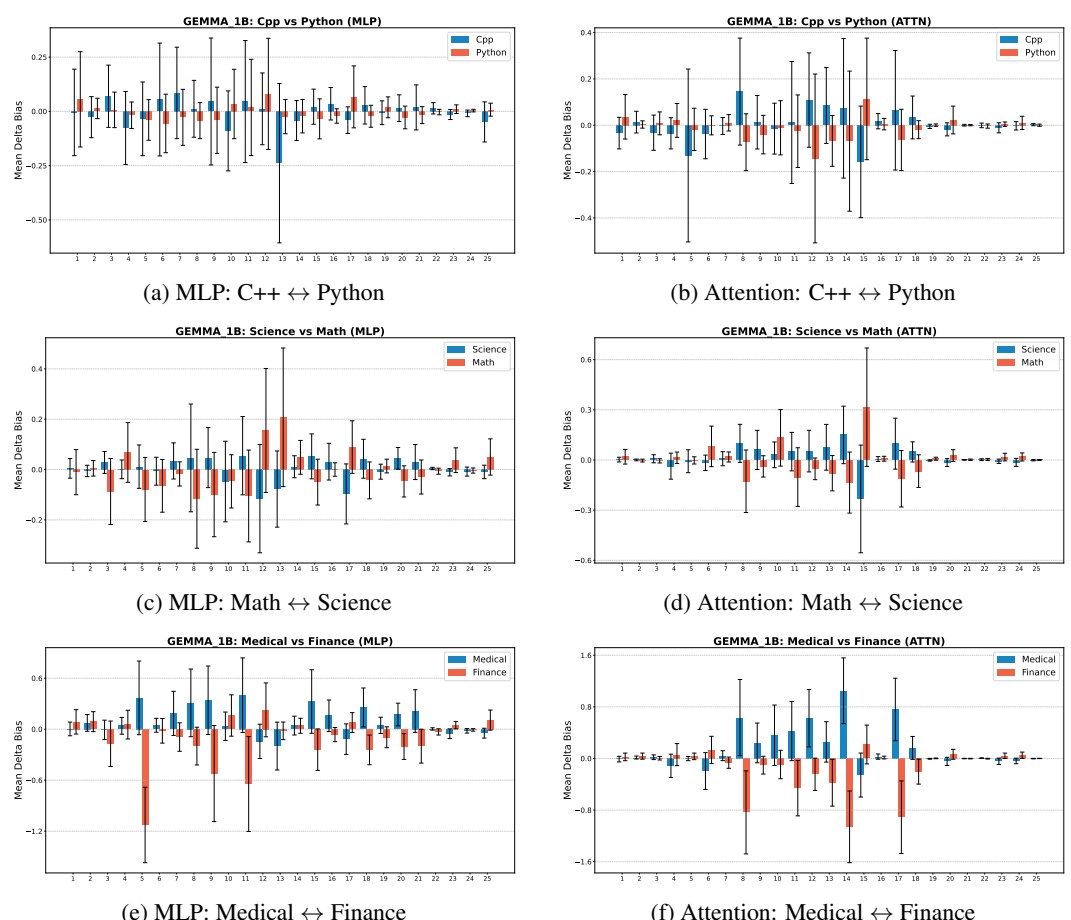

Figure 15: **Gemma-3-1B causal intervention results across all domain pairs.** Even at the smallest scale tested (1B parameters, 18 layers), the functional division persists: attention provides sparse routing, MLP provides distributed computation. The consistency across all four models (1B-4B) and three domain pairs provides robust evidence for this architectural organizing principle.

We extend our intervention study to a larger language models, such as Llama-7B, where we observe that individual layers are not highly influential in the final prediction of the token. In these layers, we observe that the earliest layers have the most impact since small changes in these layers snowball into larger changes in the final output.

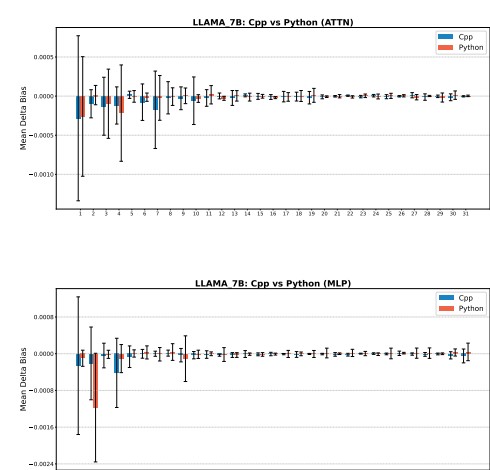

Figure 16: Results for Llama-7B showing influence of Attention vs MLP layers.

We investigated other methods of finding representative tokens from fine-tuning datasets to check if more frequent tokens are representative of a specific domain. We find that this is not the case since the even though the relative effect of the causal shift in layer swapping is still the same, the absolute magnitude in shift is reduced by %.

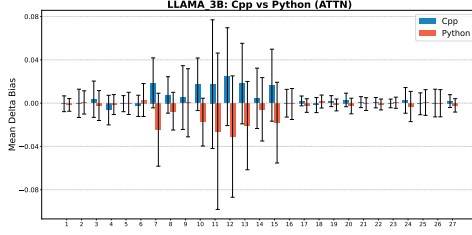

(a) Representative tokens using frequency analysis (Attention)

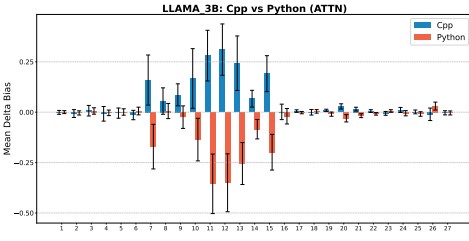

(b) LLM Generated Representative Tokens (Attention)

Figure 17: Comparison of Attention layer shifts for different token selection methods.

(a) Representative tokens using frequency analysis (MLP)

(b) LLM Generated Representative Tokens (MLP)

Figure 18: Comparison of MLP layer shifts for different token selection methods. We observe in the top graph the high variance and low quality of domain seperability.

LLM-generated tokens have been explicitly optimized for domain differentiation, whereas frequency-based tokens lack the semantic depth required for the model to distinguish between lists. This information deficit prevents effective separation, directly resulting in the higher variance observed in the top graphs.

### E.5.5  ROBUSTNESS TO TOKEN COUNT ($n$)

We tested the sensitivity of our causal results to the number of domain-representative tokens ($n$) included in each prompt list. Using the domain-classification task (Section 2.3), we varied $n \in \{5, 10, 15, 20\}$ and measured Delta Bias at the peak attention layers (13-15, 23-25).

