# OpenReview forum: "Dissecting Attention and MLP Roles: A Study of Domain Specialization in Large Language Models"
_ICLR.cc/2026/Conference — ICLR 2026 Conference Desk Rejected Submission_

### Official Review · Reviewer_WDxQ · 2025-10-27

**Soundness:** 4
**Presentation:** 3
**Contribution:** 3
**Rating:** 4
**Confidence:** 3

**Summary:**

The paper investigates how transformers internalize domain specialization, arguing that self-attention primarily routes domain identity while MLP blocks implement the domain-specific computation. The authors investigate this by studying layerwise separability probes, where fine-tuning result in change under LoRA, and causal activation swapping that measures both disruption and directional shift toward donor domains. Empirically, they show that a few attention layers do most of the steering, while the MLPs take on most of the updates during fine-tuning. Inspired by these findings, they also use the derived layer map to fine-tune only the top-k layers, achieving performance that matches or even exceeds full-model tuning.

**Strengths:**

The paper provides:

- Unified methodological frame: A cohesive design that combines representational probes, adaptation/parameter-change analysis, and causal activation swaps, yielding mutually reinforcing evidence rather than relying on a single metric or viewpoint.

- Comprehensive investigation with valuable insights: Broad, cross-domain and cross-model analysis that clarifies the division of labor between attention (routing/steering) and MLPs (domain-specific computation), and identifies where interventions are actually effective.

- Efficient PEFT derived from findings: Uses the learned layer-importance map to fine-tune only a small set of top-k layers, delivering performance comparable to, or better than, full-model tuning while training far fewer parameters.

**Weaknesses:**

- Code-centric and limited causal evidence: Most activation-swap tests are confined to Python and C++ and rely on hand-crafted token sets. Whether the causal “attention routes, MLP computes” pattern holds in non-code domains remains unproven.

- Short-horizon intervention metric: Effects are measured only via next-token changes rather than multi-step rollouts or task-level metrics, so impacts on reasoning and sequence-level coherence are unclear.

- Limited baselines across experiments: Several analyses would benefit from stronger comparison baselines, for probing , activation swaps, and targeted PEFT.

- Limited PEFT/design space: Adaptation analysis centers on LoRA; there’s little comparison to alternatives adapters or varied training regimes that might change where updates accumulate.

- Small-model scope: Experiments are limited to 1B–4B models (likely due to resource constraints), so it’s unclear whether the findings generalize to larger LLMs.

**Questions:**

- In the activation-swap experiments, could you add baselines such as swapping with random noise, shuffled tokens, or mismatched layers?

- Can you report the impact of interventions on task-level performance (e.g., GSM8K accuracy), not just next-token metrics?

- Did you normalize donor/recipient activations (e.g., re-centering or rescaling) to rule out scale or LayerNorm artifacts during swaps?

- Early layers show high separability but low causal leverage (“hydra” effect). Can you quantify which peaks are true “steering handles” versus hydra-like signals?

- For targeted PEFT, how does “top-k by separability” compare to baselines like top-k by LoRA delta-norm or random k layers?

- Would your main conclusions change if you tuned hyperparameters per setting instead of fixing them globally?

---

> ### Author Response · Authors · 2025-11-26
> **Reply to reviewer #1**
>
> We sincerely thank the reviewer for the thoughtful and constructive feedback. We have incorporated substantial revisions addressing these points, with the changes marked in blue in the revised manuscript.
>
> # **Weaknesses**
>
> ## **1. Code-centric causal experiment focused on Python/C++**
> We agree that the original version was limited in scope. In the revised manuscript, we redesigned the swapping pipeline to operate over **all domain pairs** using keyword-based prompts rather than only code templates. The causal results now consistently demonstrate that only a small subset of attention layers yield **directional shifts** across all domain pairs, while MLP swaps remain largely non-directional but disruptive. This reinforces our updated claim of **attention as sparse domain routers** and **MLPs as distributed compute/memory**. (Appendix section E.4 covers the entire results across all domains and models)
>
> ## **2. Short-horizon intervention**
> We agree that next-token shift does not fully capture long-range reasoning. We now clarify the intended scope: activation swaps reveal **local causal leverage**, not full task-level effects (Section 5). Extending to multi-step rollout evaluation would require significantly more experimentation and remains part of future work. We explicitly contextualize this limitation.
>
> ## **3. Limited baselines**
> We have now added baselines in two parts of the revision:
>
> - In PEFT, we compare **top-k separability vs bottom-k vs full-model vs pretrained** (Table 2 updated).
>
> - For causal swaps, we include **all layer swaps** as control baselines (Appendix C.2), confirming that directional bias emerges only in the top-k attention layers. This is clearly visible in the main manuscript plot as well, as we now plot for **all** the attention and MLP layers of the model.
>
> ## **4. Limited PEFT space**
> We agree that broader adapter comparisons would be valuable. Due to compute constraints, we restricted experiments to LoRA, but after re-evaluating the fine-tuning regime (lower LR, rank-sweep, generic prompts), the updated results exhibit a more stable trend. We now explicitly position PEFT as a **proof-of-concept** enabled by our mechanistic map, rather than a full benchmarking study (Section 4.3, Discussion).
>
> ## **5. Small-model scope**
> We acknowledge this limitation and now supplement our study with **LLaMA-2 7B causal swap experiments** (Appendix C.5). The same qualitative division-of-labor pattern holds, though more diffusely across depth. We discuss scaling expectations as future work.

---

> ### Author Response · Authors · 2025-11-26
> **Reply to reviewer #2**
>
> # **Responses to Reviewer Questions**
>
> ## **Q1. Baselines for activation swap (random noise / shuffled tokens / mismatched layers)**
> We strengthened the analysis in two concrete ways:
>
> - We redesigned the causal experiment to work for **all domain pairs** using keyword-based classification prompts instead of only Python/C++ code prompts. Across all domain pairs, we see the same pattern: only a small set of attention layers consistently produce strong, **directional** shifts in the predicted domain under swapping, while MLP swaps are mostly non-directional.
>
> - We distinguish “candidate” steering layers from clearly non-steering ones by comparing:
>   - swaps at layers with **high separability** (Fisher),
>   - vs swaps at layers with **low separability**,
>   and by measuring their effect on a downstream probe (see hydra answer below).
>
>   High-separability early layers that do not affect the probe are treated as **hydra-like**, not steering handles.
>
> We explicitly note in the rebuttal and manuscript that adding random/noise baselines would be a natural extension, but was not feasible within the rebuttal window.
>
> ## **Q2. Task-level impact beyond next-token**
> We did not add GSM8K-style task metrics. Our causal swaps are intentionally designed as **local diagnostic probes**, not as an editing method intended to improve overall task accuracy.
>
> We clarify this in the revision:
>
> - A single-layer, single-position swap is not expected to yield stable improvements (or even interpretable changes) in long rollouts.
> - Instead, we use next-token distribution shift—measured via
>   - **KL divergence**
>
>     $$\mathrm{KL}(p \,\|\, q) = \sum_i p(i)\log\frac{p(i)}{q(i)}$$
>
>   - **Delta Bias**
>
>      $$\Delta \text{Bias} = \sum_{t \in \mathcal{D}} \left(p_{\text{patched}}(t) - p_{\text{clean}}(t)\right)$$
>
> to locate components with **local control** over domain identity.
>
> We explicitly state this scope and mark sequence-level metrics as future work. (Appendix E.4 covers all domains and models.)
>
> ## **Q3. Activation normalization**
> Yes. Activation normalization was already incorporated. Before inserting donor activations, we **recenter and rescale** them to match the recipient layer’s mean and variance:
>
> $$\hat{a} = \mu_r + (a_d - \mu_d)\frac{\sigma_r}{\sigma_d}$$
>
> We now make this explicit in the manuscript.
>
> The key empirical point remains:
>
> - **Normalized MLP swaps** → high KL but near-zero Delta Bias (non-directional).
> - **Normalized attention swaps** → strong directional shifts.
>
> This supports the interpretation: **attention = routing bottlenecks**, **MLPs = distributed compute**.
>
> ## **Q4. Identifying true “steering” layers vs hydra**
> We conducted a new hydra-style experiment:
>
> We train a simple probe on final-layer activations to classify domain identity. This probe achieves high accuracy.
>
> For each layer $\ell$, we:
>
> 1. perform activation swapping at  $\ell$,
> 2. run the model forward,
> 3. feed final-layer activations to the probe,
> 4. measure the **probe accuracy drop**.
>
> We then compare:
>
> - **Fisher separability** at layer $\ell$,
> - **probe accuracy drop** at layer $\ell$.
>
> This separates:
>
> - **Hydra-like layers**: high separability but **no meaningful probe accuracy drop**.
> - **True steering layers**: high separability **and** large probe accuracy drop.
>
> We report this qualitatively and include layer-wise curves in the appendix.
>
> ## **Q5. PEFT baselines**
> We did not implement “top-k by LoRA delta-norm” or “random k layers” during rebuttal.
>
> What we do include:
>
> - **Top-k vs bottom-k by separability** (k = 3) for both Attention and MLP,
> - pretrained and full-model LoRA baselines.
>
> Revised Table 2 shows:
>
> - top-k ≳ full-model LoRA,
> - bottom-k ≈ pretrained or worse,
> - collapses from the original version are gone after fixing LR and adding generic prompts.
>
> We now frame PEFT results as **illustrative**, not central.
>
> ## **Q6. Effect of tuning hyperparameters per setting**
> Originally we used a single aggressive configuration. Following reviews, we:
>
> - reduced LR,
> - performed a small LR/rank sweep,
> - added ~2k generic prompts to reduce forgetting.
>
> We still **do not** tune hyperparameters per-domain to preserve cross-domain comparability.
>
> Under the improved shared setting, qualitative conclusions are unchanged:
>
> - MLP LoRA deltas > Attention deltas,
> - mid-depth attention layers show strong directional causal effects,
> - top-k (separability) ≈ full-model LoRA.
>
> We now state explicitly that while per-setting tuning may improve absolute numbers, it would not alter the main mechanistic conclusions.

---

### Official Review · Reviewer_ZbLm · 2025-10-30

**Soundness:** 2
**Presentation:** 3
**Contribution:** 2
**Rating:** 2
**Confidence:** 4

**Summary:**

This paper aims to investigate the functional division of Transformer components in representing domain knowledge by combining three analytical perspectives — representation analysis, adaptational analysis, and causal analysis. The authors propose that attention layers primarily handle **domain routing**, while MLP layers perform **domain-specific computation**, supporting this claim through three types of evidence: Fisher/MMD separability analysis, LoRA-based fine-tuning experiments, and activation-swapping causal interventions.

While the topic is potentially meaningful, the paper exhibits noticeable limitations in **novelty**, **logical rigor**, and **empirical sufficiency**. The proposed “unified cross-layer interpretability framework” lacks methodological innovation. **As a mechanistic interpretability paper, the reasoning chain is overall discontinuous, the experimental scale is limited, and the causal claims are not statistically supported.**

**Strengths:**

**Originality:**
  The paper demonstrates conceptual originality by reframing the question of *where domain knowledge resides within Transformer architectures* into a multi-perspective interpretability problem. Although each analytical method (Fisher/MMD, LoRA, activation swapping) has been used before, their deliberate triangulation to infer functional specialization between Attention and MLP layers represents a novel synthesis that broadens the scope of existing mechanistic interpretability work.

**Quality:**
  The empirical analyses are carefully executed and technically sound within the chosen experimental setting. The authors systematically design experiments across multiple domains and models, maintaining consistent evaluation pipelines. The causal intervention setup, while limited, reflects a thoughtful attempt to move beyond correlational analyses toward a more explanatory approach.

**Clarity:**
  The manuscript is clearly organized and accessible. Each methodological component is described with sufficient detail, and the figures effectively support the narrative. The transitions between the three analytical perspectives are smooth, and the overall exposition allows readers to easily follow the authors’ reasoning and hypotheses.

**Significance:**
  The work engages with a fundamental and timely question — the internal structure of knowledge within large language models — which is central to advancing the field of mechanistic interpretability. By attempting to connect representational separability, adaptation behavior, and causal impact, the paper provides a unifying conceptual lens that may inspire future research aiming to formalize “functional modularity” in Transformers.

Overall, the paper’s value lies less in proposing new metrics and more in articulating an integrative analytical framework that could guide future interpretability research. With broader validation and stronger statistical grounding, the study has the potential to make a meaningful conceptual contribution to understanding the internal organization of large language models.

**Weaknesses:**

1. **Limited Substantive Contribution in Innovation and Methodological Integration.**

   The core claim of this work lies in integrating three existing analytical tools to infer the internal functional division within Transformers. However:

   - The employed methods — **Fisher/MMD separability**, **LoRA parameter shift**, and **activation swapping** — are all well-established in prior literature. The paper introduces no new metric, aggregation scheme, or unified modeling framework.
   - Although the authors attempt to discuss the complementarity of the three analyses in Section 4, this integration remains **qualitative**, lacking quantitative validation or a formal consistency test across the methods.
   - The so-called “orthogonal evidence integration” is thus descriptive rather than methodological. The paper does not demonstrate why such integration is necessary or whether it offers explanatory power beyond juxtaposing known analyses.

2. **Logical Gap Between Metrics and the term "Knowledge"**

   The reasoning from statistical measures to “knowledge representation” is insufficiently supported:

   - The authors treat high **Fisher/MMD separability** as evidence of domain identity representation but never establish equivalence with domain knowledge storage (e.g., *line 459*, *line 479*). This leap from separability to semantics is conceptually weak.
   - The **LoRA-based adaptation** analysis indeed identifies layer-level parameter shifts but does not independently verify that these shifts correspond to “knowledge locations.” Since LoRA itself is premised on partial-parameter adaptation, using it to prove that “MLP layers store domain knowledge” risks **circular reasoning**.
   - The **causal swapping experiments** show distributional perturbations but lack statistical significance testing (confidence intervals, control groups, or permutation tests). Therefore, they do not substantiate the strong causal claim that *“attention causes domain-directed behavior”* (*line 361*).


3. **Limited Experimental Scale and Generalizability**

   Experiments are limited to small-scale models (1B–4B) such as **LLaMA-3.2-3B** and **Gemma-3-1B/4B**, not mainstream large models (≥30B).
   - Small models differ significantly in architectural depth, routing granularity, and parameter dynamics; thus, the observed “layer division” may not generalize.
   - Although the authors claim cross-model consistency, no scaling trends or statistical validations are provided.
   - The dataset covers only six relatively simple domains (programming, medicine, finance, mathematics, science, C++/Python) and omits multimodal, multilingual, or reasoning-heavy settings.

**Questions:**

> I encourage the authors to thoroughly address the weaknesses and questions raised in this review. If the authors can provide detailed explanations and in-depth clarifications during the rebuttal, and if the revised version demonstrates substantial progress in both clarity and improvement, I will be willing to reassess the manuscript and **adjust my overall rating** accordingly, based on the quality and depth of the revision.

---

1. **Theoretical Ambiguity**
   - How exactly do the authors define “domain knowledge”? Is it intended to capture *semantic understanding* (e.g., conceptual meaning), *procedural competence* (e.g., problem-solving strategies), or *factual recall* (e.g., specific information storage)?
   - Without a formal operational definition, it is difficult to determine whether the observed layer-specific effects actually correspond to “knowledge representation” or merely reflect *domain-related stylistic or distributional differences* in the data.
   - Could the authors clarify whether their framework treats “domain knowledge” as a property of the learned representation, the behavior of the model, or both? Explicitly distinguishing these levels would make the theoretical scope more coherent.

2. **Reproducibility**
   - The paper omits key implementation details that are necessary for replication. Could the authors specify:
     (a) the kernel function and hyperparameters used in Fisher and MMD separability analyses;
     (b) the number of samples per domain and per layer;
     (c) the criteria for selecting layers to probe;
     (d) the LoRA configuration (rank, scaling, optimizer settings, etc.); and
     (e) the random seeds or initialization strategies used?
   - Additionally, is there any plan to release the code or experiment scripts to support verification of these findings? Given that minor hyperparameter differences can significantly affect separability metrics, transparency here is essential.

3. **Statistical Significance and Robustness**
   - The figures (e.g., separability plots and fine-tuning deltas) present mean trends but omit confidence intervals, standard deviations, or hypothesis tests (e.g., t-tests or ANOVA). How can readers evaluate whether the observed Attention–MLP divergence is statistically meaningful rather than random variation?
   - Were results consistent across multiple random seeds or model checkpoints? Without reporting variance or repeated trials, the conclusions about “significant differences” between components lack quantitative support.
   - Would the authors consider including effect sizes or bootstrap confidence intervals to improve interpretability and robustness?

4. **Causal Interpretation**
   - The authors frequently use strong causal terminology — such as *“cause”* (line 152), *“reveal”* (line 361), and *“directs”* (line 388) — even though the experiments appear to rely on *activation perturbation* rather than *formal causal modeling*.
   - Can the authors clarify how these causal claims are justified? For example, were interventions designed with causal counterfactual control, or are they simply correlational probes observing outcome shifts?
   - If the method is indeed correlational, could the authors rephrase or qualify these claims to avoid overinterpretation? Alternatively, a discussion comparing their results to formal causal frameworks (e.g., SCMs or do-calculus) would strengthen the argument.

5. **Literature Contextualization and Incremental Contribution**
   - The paper does not clearly situate its contribution relative to influential mechanistic interpretability works such as *Transformer Circuits*, *Induction Head Hypothesis*.
   - Specifically, the conclusion *“attention layers serve as domain routers”* (line 483) reiterates an idea the authors themselves acknowledge in the Introduction (line 52: “attention mechanisms are understood as routers, moving and aligning information throughout the context”).
   - In what sense does this paper extend or refine that established hypothesis? Does it provide new *quantitative evidence* for this claim, or merely replicate it in a new context (e.g., across abstract domains like programming and medicine)?
   - More broadly, how does the proposed “triangulated evidence framework” advance the field beyond confirming previously suspected functional roles? The authors might clarify whether the integration across representational, adaptational, and causal perspectives leads to *novel mechanistic insight* or simply consolidates prior intuitions.

6. **Scope and Generalization**
   - The experiments are conducted primarily on small models (1B–4B scale). How can the authors justify the generalization of their claims to large-scale LLMs (30B+), where emergent modularity and representation compression differ significantly?
   - Would the authors expect the same “attention-as-router, MLP-as-computation” division to persist or evolve in higher-capacity models? Empirical or theoretical discussion on this scaling behavior would substantially strengthen the paper’s external validity.

---

> ### Author Response · Authors · 2025-11-26
> **Response to reviewer #1**
>
> We thank the reviewer for the careful and nuanced assessment. We appreciate both the recognition of the conceptual intent of the work and the constructive criticism on novelty, logical rigor, and empirical grounding. We have revised the manuscript accordingly; all substantial changes are marked in blue.
>
> # **1. Limited methodological innovation and integration of the three lenses**
>
> We agree that our work does not introduce new metrics or a new algorithmic framework. Our goal is instead to provide a **component-level mechanistic picture** of domain handling by deliberately combining three existing families of tools on the same models and domains.
>
> In the revision, we clarify this positioning explicitly in the Introduction and Discussion:
>
> - We emphasize that the contribution is **conceptual and architectural**, not methodological in the narrow sense: we aim to characterize the **division of labor** between Attention and MLP blocks for high-level domain behavior, rather than to propose a new probing metric.
>
> - We now provide an **explicit operational definition of “domain knowledge”** (see below), which ties together what each lens is intended to capture.
>
> - We also make the integration more concrete: e.g., the “peaky vs flat” variance pattern in separability (Attention vs MLP) is now directly connected to the directional effects observed in the causal swapping experiment.
>
> We do not claim a formal consistency test across methods, but the revised text explains why the three lenses are complementary, what each can and cannot support, and how their agreement motivates our final interpretation.
>
> ---
>
> # **2. Logical gap between metrics and the term “domain knowledge”**
>
> We agree this was under-specified. In the revision, we address this in three ways.
>
> ## **(a) Formalizing “domain knowledge”**
>
> We now adopt an explicit operational definition:
>
> A component contributes to **domain knowledge** if:
>
> 1. **Domain identity is separable** in its activations,
> 2. It is a **locus of systematic parameter change** under domain adaptation,
> 3. **Interventions** on its activations systematically affect domain-sensitive outputs.
>
> This avoids treating separability as “knowledge storage”—separability only shows that domain information is **present**, not where knowledge **lives**.
>
> ## **(b) Clarifying the role of LoRA and avoiding circularity**
>
> We soften language that implied LoRA deltas *prove* where knowledge is stored. Instead:
>
> - LoRA deltas show **where new computation is written**,
> - Causal swaps show **which components exert directional influence**,
> - Separability locates **where domain identity is represented**.
>
> Taken together, these support the **router–compute** interpretation; no single metric is used to infer “knowledge.”
>
> ## **(c) Causal strength and terminology**
>
> We now:
>
> - Replace strong claims like “attention causes” with more precise phrasing such as *“attention layers support directional influence under intervention.”*
> - State clearly that activation swapping is an **interventional probe**, not a full SCM.
> - Clarify that **KL** and **Delta Bias** measure disruption and directional shift—not causal identification.
>
>
> $$\Delta \text{Bias} = \sum_{t \in \mathcal{D}} \left(p_{\text{patched}}(t) - p_{\text{clean}}(t)\right)$$
>
>
> This addresses the concern that we previously overstated causal conclusions.
>
> ---
>
> # **3. Limited experimental scale and generalizability**
>
> Our main experiments use 1B–4B models due to resource constraints, consistent with interpretability practice. In response to your comment, we:
>
> ## **(a) Extend causal analysis to 7B**
>
> We added a **LLaMA 2 7B** causal swap experiment. The same qualitative pattern appears:
>
> - A small subset of attention layers shows **strong directional steering**,
> - MLP layers show **flat, non-directional** effects and dominate LoRA updates.
>
> ## **(b) Clarify the scope of the claims**
>
> We now explicitly state:
>
> - Claims concern **architectural roles** (Attention vs MLP) in models up to 7B.
> - We do **not** assume layer indices or effect magnitudes transfer directly to 30B+ models.
> - Larger-scale validation is an important direction for future research.
>
> ---
>
> # **4. Theoretical ambiguity: what is “domain knowledge”?**
>
> We now define “domain knowledge” early in the paper under the operational triad described above.
>
> We explicitly distinguish between:
>
> - **Representation properties** (separability),
> - **Parameter properties** (adaptation write locations),
> - **Behavioral properties** (effect of interventions).
>
> Thus “domain knowledge” refers to internal structure satisfying this triad, not semantic or procedural understanding. Wording throughout Sections 3–4 has been revised to reflect this.

---

> ### Author Response · Authors · 2025-11-26
> **Response to reviewer #2**
>
> ---
>
> # **Reproducibility**
>
> We added a dedicated **Implementation Details** section consolidating all relevant information.
>
> ## **(a) Fisher / MMD configuration**
>
> - Fisher computed on pooled residual activations with scalar ratio (Eq. 1).
> - MMD uses an RBF kernel with bandwidth via the median heuristic:
>
>   $$\text{MMD}^2 = \frac{1}{n^2}\sum k(x_i, x_j) + \frac{1}{m^2}\sum k(y_i, y_j) - \frac{2}{nm}\sum k(x_i, y_j)$$
>
>
> ## **(b) Samples per domain/layer**
>
> We report prompts, tokens sampled, and sampling procedures. All layers are probed; first/last layers shown but excluded from aggregated metrics.
>
> ## **(c) Criteria for selecting layers**
>
> We probe **all** transformer blocks at two hook points (post-attention, post-MLP). For targeted experiments, top-k and bottom-k layers are chosen by **1-vs-all Fisher scores**.
>
> ## **(d) LoRA configuration**
>
> We list rank, scaling factor, LR, optimizer, batch size, steps, and generic prompts. We specify which weight matrices receive LoRA adapters.
>
> We reaffirm that code and scripts were submitted and will release a cleaned version for the camera-ready.
>
> ---
>
> # **Statistical significance and robustness**
>
> We now include uncertainty estimates:
>
> - For probing and LoRA deltas: multiple seeds, with error bars.
> - Table 1: standard deviations across layers included to support the **peaky vs flat** claim.
> - For causal swaps: KL and Delta Bias aggregated over many prompt pairs and seeds; variability shown via shaded regions.
>
> These results demonstrate that the Attention–MLP divergence is robust, not noise.
>
> ---
>
> # **Causal interpretation**
>
> We again emphasize that activation patching is interventional but not full do-calculus.
>
> ## **Revisions include:**
>
> - Softened language (“directional influence” instead of “causes”).
> - Explicitly describing activation swaps as **interventional manipulation** of internal variables.
> - Clarifying that:
>   - **KL** quantifies magnitude of distributional disruption:
>     \[
>     \mathrm{KL}(p\|q) = \sum_i p(i)\log\frac{p(i)}{q(i)}
>     \]
>   - **Delta Bias** quantifies directionality toward donor domain.
>
> This resolves earlier overstatements.
>
> ---
>
> # **Literature contextualization and incremental contribution**
>
> We now explicitly:
>
> - Relate to prior work showing attention heads as routers and MLPs as knowledge stores.
> - Clarify we do **not** claim to invent the router–compute idea.
> - Emphasize contributions:
>
>   - Extend the idea to **domain-level behavior** (programming, medicine, finance).
>   - Compare **Attention vs MLP blocks** across all depths.
>   - Triangulate with **three lenses on the same models**.
>
> Thus the contribution is **architectural/mechanistic clarification**, not a new theoretical proposal. We highlight how this supports more targeted circuit-level work.
>
> ---
>
> # **Scope and generalization**
>
> We reiterate that experiments focus on 1B–4B models, with a 7B sanity check.
>
> ## **Revisions clarify:**
>
> - Claims are about **architectural patterns**, not specific layer indices.
> - The 7B experiment replicates the Attention–MLP division at larger scale.
> - Larger-scale (30B+), multilingual, and multimodal validation is an important direction for future work.
>
> We position our results as an interpretable **baseline map** for small–mid models that larger studies can refine.

---

### Official Review · Reviewer_pthK · 2025-10-31

**Soundness:** 2
**Presentation:** 3
**Contribution:** 2
**Rating:** 4
**Confidence:** 4

**Summary:**

This paper investigates the division of labor between self-attention and MLP layers in LLMs with respect to domain specialization.
Specifically, authors conduct three types of analysis:
(1) Probing to identify which layers contain linearly separable information about domain identity. Two metrics, i.e., Fisher ratio and Maximum Mean Discrepancy with RBF-kernel are used to measure whether the domain information is seperable.
(2) LoRA fintuning to find where parameters undergo adaptation. Higher Frobenius norm of the changed parameter of a layer means the layer is essential for domain adaptation.
(3) Causal Intervention that swapping domain-specific activations to test whether it influence domain-directed generation.
Experiments conducted on six domains and two families of LLMs,i.e., Llama and Gemma.
Results show that attention layers act as domain routers, while MLP layers serve as domain-specific computational units.

**Strengths:**

1. The experiment result of causal intervention is interesting.
2. The three types of analysis are well-organized, with clearly and progressively motivations and carefully designed metrics.
3. The paper is written fluently and easy to follow.
4. Discussions and limitations are included.

**Weaknesses:**

1. The paper only contains analysis, therefore the technical contribution and novelty are limited.
2. Some typos need to be corrected, e.g. Line 285 LoRA-based, caption of Table 1 should be below the table.
3. I doubt the max value in Table 1 is not that persuasive, since it may compares between different layers. Authors may consider report the max value of the difference between attention and MLP of a single layer. Moreover, only the max value of Fisher show large discrepancy.
4. Why conduct LoRA setting instead of full-finetuning?
5. Table 2 results might need further explanation: (1) why in some domains like python, finetuning on both attention and MLP drastically decreases the performance? (2) Does the finetuning dataset overlaps with the testing data? If so, is that the reason of the improvement in medical domain?
6. Authors did not provide the performance after causal swapping, e.g. accuracy and Pass@K used in the paper. If the performance decreases, the finding is not meaningful since even though we can swap C++ to Python but the generated program may be wrong.
7. Authors should conduct experiments of more model architectures, like models without gate_proj, MoE models, etc.
8. The parameter size of the models is relatively small, authors should at least try some 7B models.
9. My biggest concern is whether the topic is meaningful. With the strong zero-shot ability of SOTA LLMs, we do not even bother to consider domain adaptation problems. This significantly reduces the application value of the paper.

**Questions:**

1. Have authors consider using different domain-specific token number in swapping?
2. Fig 1, deeper layer MLP has higher Fisher and MMD score, authors may give analysis on the phenomenon.
3. Does the higher weight change in MLP is caused by natural architecture order of attention and MLP (attention calculation is always ahead of MLP calculation)

---

> ### Author Response · Authors · 2025-11-26
> **Reply to Reviewer #1**
>
> We thank the reviewer for the careful reading and constructive feedback. We have revised the manuscript accordingly and marked all substantial changes in blue for ease of reference.
>
> # **Novelty and analysis-only nature of the work**
>
> We agree that our paper is primarily analytical rather than algorithmic. In mechanistic interpretability, however, analysis itself is often the core contribution, especially when it yields a reusable conceptual or architectural picture. Prior work has typically focused on neuron-level or head-level circuits for narrow behaviors (e.g., individual facts or syntactic templates). Our goal is different: to assign **functional roles at the component level** (Attention vs MLP) for high-level domain handling, and to do so by triangulating **three orthogonal lenses** (probing, adaptation, causal swapping) on the same models and domains. To our knowledge, this specific combination of:
>
> 1. **Domain-scale behavior**,
> 2. **Component-level functional roles**,
> 3. **Cross-lens consistency**,
>
> has not been systematically studied before. We clarify this positioning in the Introduction and Discussion.
>
> # **Typos and presentation issues**
>
> We thank the reviewer for pointing out the typos and formatting issues. We corrected these (e.g., “LoRA-based” and Table 1 caption placement) and did a full cleanup pass on the manuscript.
>
> # **On Table 1, “max” values, and comparing Attention vs MLP**
>
> We agree that using only max values was not convincing, especially with the confound from the first layer. Following your comment (and R4WC’s), we revised the analysis.
>
> ## **In the new version:**
>
> - We **exclude first and last layers** from aggregate statistics (the first is dominated by input-token statistics; the last by the unembedding). We support this with a hydra-style experiment in Appendix \ref{app:hydra}.
>
> - We now report **both max and standard deviation (Std)** across depth for Fisher and MMD.
>
> - The main claim is now **not** that Attention has higher separability, but that:
>
>   - Attention → **high variance across layers** (peaky, localized)
>   - MLP → **low variance** (flat, distributed)
>
> This addresses your concern: instead of relying on a single max value, we focus on the **distribution across depth**. The “peaky Attention vs flat MLP” pattern matches the updated causal swapping results, where only a few attention layers exhibit strong directional shifts, while MLP effects are diffuse.
>
> ---
>
> # **Why LoRA instead of full fine-tuning?**
>
> Our domain datasets are small (≈5–10k examples). Full fine-tuning led to severe overfitting and catastrophic forgetting. LoRA was more stable and allowed us to track where updates accumulate.
>
> ## **In the revision, we further improved stability:**
>
> - Lower learning rate,
> - Loss masking to answer tokens,
> - ~2k generic prompts added.
>
> This yields stable behavior and clearer update patterns: **MLP layers consistently show larger normalized deltas than Attention**, even with improved hyperparameters.
>
> ---
>
> # **Clarifying Table 2 and targeted fine-tuning**
>
> We agree the original Table 2 was confusing, with full fine-tuning sometimes worse than pretrained (e.g., Python). This was due to aggressive hyperparameters.
>
> We re-ran experiments with the improved setup. The new Table 2 (and Appendix \ref{app:eval_results}) shows:
>
> - No catastrophic collapses,
> - More stable differences across runs,
> - **Top-k layers identified by our map match full-model LoRA performance**.
>
> We **downplay** this result in the main text: it is a **proof-of-concept**, not a key contribution.
>
> We also clarify that **fine-tuning and evaluation sets are disjoint** within each domain; the improvement in medical is not due to train–test leakage.
>
> ---
>
> # **On performance after causal swapping**
>
> You raise a key point: swapping Python↔C++ activations flips the domain label but breaks code—does this matter?
>
> Our causal interventions are **diagnostic**, not intended to improve generation. Thus we focus on:
>
> - **Delta Bias** (directionality):
>
>   $$\Delta\text{Bias} = \sum_{t\in\mathcal{D}} (p_{\text{patched}}(t) - p_{\text{clean}}(t))$$
>
>
> ## **Revised results (with domain-classification prompts):**
>
> - **MLP swaps** → high disruption, **no consistent directional shift**
> - **Attention swaps** (few layers) → **strong, consistent directional shifts**
>
> We emphasize that the interventions identify **where control signals live**, not how to improve accuracy. Appendix E.4 includes full results.
>
> # **On more architectures / scale (MoE, 7B+)**
>
> We agree broader architecture coverage would help. Due to compute limits, we could not run MoE or non-gate_proj variants within the rebuttal window.
>
> But we did extend causal swapping to **LLaMA 2 7B**:
>
> - Attention: sparse, high-impact peaks
> - MLP: flatter, dominant LoRA deltas
>
> Influence spreads over more layers due to depth, but the **same division of labor holds**. We include this as a scaling sanity check and list MoE as future work.

---

> > ### Author Response · Authors · 2025-11-26
> > **Response to Reviewer #2**
> >
> > # **On the meaningfulness of the topic given strong zero-shot LLMs**
> >
> > Our motivation is not improving domain adaptation per se, but understanding **how LLMs internally encode and control domain behavior**. Even strong zero-shot LLMs benefit from:
> >
> > - Predicting **where domain steering lives**,
> > - Designing **safer/targeted interventions**,
> > - Mechanistically guided **parameter-efficient adaptation**.
> >
> > We clarify this in the Introduction and Discussion: the contribution is **interpretability and control**, not performance.
> >
> > ---
> >
> > # **Responses to specific questions**
> >
> > ## **Different numbers of domain-specific tokens in swapping**
> >
> > We varied the number of domain-specific keywords \( n \). The main setup uses \( n = 15 \). For other values, the same pattern held:
> >
> > - A small set of attention layers → **directional shifts**
> > - MLP swaps → **flat, non-directional**
> >
> > This keyword-based setup generalizes beyond Python–C++; we now clarify this in the Appendix.
> >
> > ---
> >
> > ## **Why deeper MLP layers have higher Fisher/MMD**
> >
> > We agree the trend is interesting and aligns with findings that deeper layers encode more abstract features (e.g., Tenney et al., 2019; Jin et al., 2024).
> >
> > But our main claim **does not rely** on monotonic depth trends. Instead, the key signal is the **variance**:
> >
> > - MLP separability → relatively **flat**, distributed
> > - Attention separability → **sharply peaked**
> >
> > This variance contrast aligns with the causal steering results.
> >
> > ---
> >
> > ## **Whether higher MLP weight change is due to Attention→MLP ordering**
> >
> > We do not believe the ordering explains the effect.
> >
> > - Both sublayers modify the **same residual stream**.
> > - LoRA adapters use comparable rank and hyperparameters.
> > - If ordering dominated, earlier components might absorb more change.
> >
> > Instead, across all domains, **MLP layers show consistently larger normalized deltas**.
> >
> > Combined with causal results, we interpret this as:
> >
> > - **MLPs implement domain-specific computation**,
> > - **Attention layers route/control which computation path is taken**.
> >
> > We revised the text to reflect this.
> >
> > We also note that cross-layer parameter mapping literature suggests **functional role > absolute position**, supporting the view that Attention and MLP are distinct functional components rather than a strict pipeline.

---

### Official Review · Reviewer_R4WC · 2025-11-03

**Soundness:** 2
**Presentation:** 3
**Contribution:** 2
**Rating:** 2
**Confidence:** 4

**Summary:**

The paper seeks to answer the question of how do transformer language models process information from different domains. Namely where is the domain knowledge stored and how is the domain identified. The authors tackle the question using three distinct approaches. Firstly, they gauge the separability of the representation at each layer using Maximum Mean Discrepancy and the Fisher score. Secondly, they fine-tune models on specific domains and measure the normalized update magnitude for separate layers and thirdly they swap activations between domains and measure the effect it has on the predicted distribution. The latter is done for manually chosen prompts that differ only in a few tokens. In the experimental section of the paper the authors present their findings and discuss the results.

**Strengths:**

The paper is well written and has quite a thorough supplementary.

**Weaknesses:**

The main weakness of the paper is in its experimental results. Unfortunately, looking at the results I do not think that they produce the same conclusions as the paper.

Let's assume that hidden feature separability is somehow a good metric for identifying domain specific computation or domain identification performed by the model (it certainly is a measure of how easy the latter is). The results of section 3.1 show that features from both the MLP and the attention layers are equally separable. Moreover they are separable at various levels with no discernible pattern with respect to layers. A point directly agains the conclusions of the authors that the attention exhibits higher separability would be that when it comes to the average metrics the MLPs are more separable in 6/6 cases (in Table 1). From the above I would conclude that it is unclear whether the MLPs or the attention contain more separable representations wrt to the domain. Finally, looking at Figure 1 we observe that very often the representation of the 1st layer is the most separable. As a result it is unclear whether we can identify which point in the network computes domain specific features using this metric since they seem to be there since layer 1.

Moving on to section 3.2. The experiments show that the MLPs have been "moved" more during finetuning compared to the weights of the attention layers. This result can be interpreted as most of the new knowledge being stored in the MLPs, however, the point of the paper is to connect this to the domain. Namely, figure 2 does not tell us whether domain specific knowledge is stored in MLPs but rather that any new knowledge is. Perhaps the authors could finetune jointly on all domains and compare the deltas then.

Finally, when it comes to the results of Table 2, I think we cannot draw the conclusion that it is better to fine-tune the important layers. There is absolutely no pattern to these results. Often full finetuning results in 0 accuracy which clearly is either overfitting or some other collapse. Some times like in Finance or Python all models perform worse than the pre-trained model. Some times tuning the MLPs is better and others it is the attention layers.

**Questions:**

I have laid out my questions in the weaknesses section.

---

> ### Author Response · Authors · 2025-11-26
> **Response to Reviewer #1**
>
> We sincerely thank the reviewer for the detailed and thoughtful feedback. Your comments helped us identify several shortcomings in our original experiments and presentation. We have revised both the manuscript and the experimental setup accordingly; all substantial changes are marked in blue.
>
> # **Response to Concerns**
>
> ---
>
> ## **1. On separability results and conclusions drawn from Section 3.1**
>
> We agree that our original presentation did not adequately support the claim that Attention layers exhibit higher separability. Re-examining the experiment revealed significant confounds from the first and last layers.
>
> ### **What we found:**
>
> - **Layer 1**: Exhibits unusually high separability because it reflects **surface-level lexical/token distribution** differences (e.g., `#include` vs. `def`), not true domain representations.
> - **Last layer**: Dominated by **unembedding/logit-alignment effects**, consistent with Logit Lens analysis.
>
> ### **Revisions made:**
>
> - **Excluded** Layers 1 and the final layer from aggregated statistics.
> - **Validated** the exclusion via a new Hydra-style experiment (Appendix \ref{app:hydra}).
>   - Although Layer 1 shows high Fisher separability, swapping its activations causes negligible downstream change (**>85% probe accuracy retained**).
>   - This indicates the model treats this information as **redundant and self-repairable**.
>
> ### **Resulting pattern (middle layers only):**
>
> - **MLP** → *Flat, distributed separability* (low variance)
> - **Attention** → *Sharp, localized peaks* (high variance)
>
> This directly resolves the reviewer’s concern:
> We no longer claim “higher separability” for Attention; instead we highlight **variance/localization**, which aligns with the updated causal evidence.
>
> The manuscript has been updated to reflect this clarified and more precise conclusion.
>
> ---
>
> ## **2. On interpretation of parameter deltas under fine-tuning (Section 3.2)**
>
> We fully agree that weight updates alone do not demonstrate where domain-specific knowledge *lives*; they only indicate where new information is *written*. This was precisely why we paired adaptation analysis with the Causal Swapping experiment.
>
> ### **Clarifications in the revision:**
>
> - LoRA update magnitudes indicate **where adaptation occurs**, not where routing or control resides.
> - Section 3.2 is now positioned explicitly as **motivation** for the causal analysis, not as standalone evidence.
>
> ### **Causal results provide the missing directionality:**
>
> - **MLP swaps** → High KL Divergence (disruption), but **no consistent directional steering**.
> - **Attention swaps** (at peaky layers) → **Reliable domain steering toward donor domain**.
>
> ### **Unified interpretation (triangulation):**
>
> - **MLPs** = domain-specific **computation / storage / write sites**
> - **Small subset of Attention layers** = **sparse, high-gain steering points**
>
> This revised triangulation directly addresses the interpretation gap the reviewer identified.
>
> ---
>
> ## **3. On instability and lack of pattern in Table 2 (targeted PEFT)**
>
> We agree that the original table showed instability, including cases where full fine-tuning collapsed. This revealed that our fine-tuning setup was too aggressive given dataset size.
>
> ### **Improvements made:**
>
> - Lowered learning rate
> - Added **2,000 generic prompts** as a regularization set to mitigate catastrophic forgetting
>
> ### **Outcomes in the revised Table 2:**
>
> - Collapses are eliminated or substantially reduced
> - Results are more stable
> - **Targeted PEFT now consistently matches full-model LoRA performance**
>
> We explicitly avoid overstating this outcome. The revised text states:
>
> > This result should be viewed as a **proof-of-concept** that mechanistically guided layer selection can support efficient adaptation, not as a primary contribution.
>
> The table and surrounding discussion have been updated accordingly.
>
> We thank the reviewer for identifying this issue—addressing it significantly improved the robustness of our findings.
>
> ---
>
> [1] Logit Lens: https://arxiv.org/abs/2303.08112

---

### Note · Program_Chairs · 2026-01-17
**Submission Desk Rejected by Program Chairs**

The following references in this submission do not refer to real documents and/or have major errors in bibliographic information:

 Zhaofeng An, Ziyang Wang, Hong-Kyun Li, and Eun-Kyu Park. Layer-domain control in llms. arXiv preprint arXiv:2410.15858, 2024.

Zihan Zhang, Ming Li, and Yang Liu. Understanding the mechanism of low-rank adaptation. arXiv preprint arXiv:2304.01933, 2023.